# Dextr: Zero-Shot Neural Architecture Search with Singular Value Decomposition and Extrinsic Curvature

**Rohan Asthana**                                         *rohan.asthana@fau.de*
*Friedrich-Alexander-Universität Erlangen-Nürnberg*
*Erlangen, Germany*

**Joschua Conrad**                                        *joschua.conrad@uni-ulm.de*
*Universität Ulm*
*Ulm, Germany*

**Maurits Ortmanns**                                      *maurits.ortmanns@uni-ulm.de*
*Universität Ulm*
*Ulm, Germany*

**Vasileios Belagiannis**                                 *vasileios.belagiannis@fau.de*
*Friedrich-Alexander-Universität Erlangen-Nürnberg*
*Erlangen, Germany*

**Reviewed on OpenReview:** *https://openreview.net/forum?id=X0vPof5DVh*

## Abstract

Zero-shot Neural Architecture Search (NAS) typically optimises the architecture search process by exploiting the network or gradient properties at initialisation through zero-cost proxies. The existing proxies often rely on labelled data, which is usually unavailable in real-world settings. Furthermore, the majority of the current methods focus either on optimising the convergence and generalisation attributes or solely on the expressivity of the network architectures. To address both limitations, we first demonstrate how channel collinearity affects the convergence and generalisation properties of a neural network. Then, by incorporating the convergence, generalisation and expressivity in one approach, we propose a zero-cost proxy that omits the requirement of labelled data for its computation. In particular, we leverage the Singular Value Decomposition (SVD) of the neural network layer features and the extrinsic curvature of the network output to design our proxy. Our approach enables accurate prediction of network performance on test data using only a single label-free data sample. Our extensive evaluation includes a total of six experiments, including the Convolutional Neural Network (CNN) search space, i.e. DARTS and the Transformer search space, i.e. AutoFormer. The proposed proxy demonstrates a superior performance on multiple correlation benchmarks, including NAS-Bench-101, NAS-Bench-201, and TransNAS-Bench-101-micro; as well as on the NAS task within the DARTS and the AutoFormer search space, all while being notably efficient. The code is available at https://github.com/rohanasthana/Dextr.

## 1 Introduction

Following the success of standard deep neural network architectures (He et al., 2016; Szegedy et al., 2015; LeCun et al., 1998; Vaswani et al., 2017), the search for optimal network topologies, referred to as Neural Architecture Search (NAS), has gained significant attention in the past years. Search-based (Li & Talwalkar,

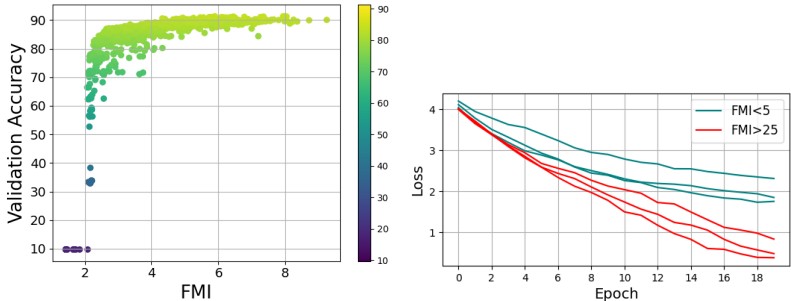

(a) (left) Scatter plot between the network performance and linear independence of feature maps (FMI). (right) Training loss curve for six networks sampled from NAS-Bench-201 on CIFAR-10.

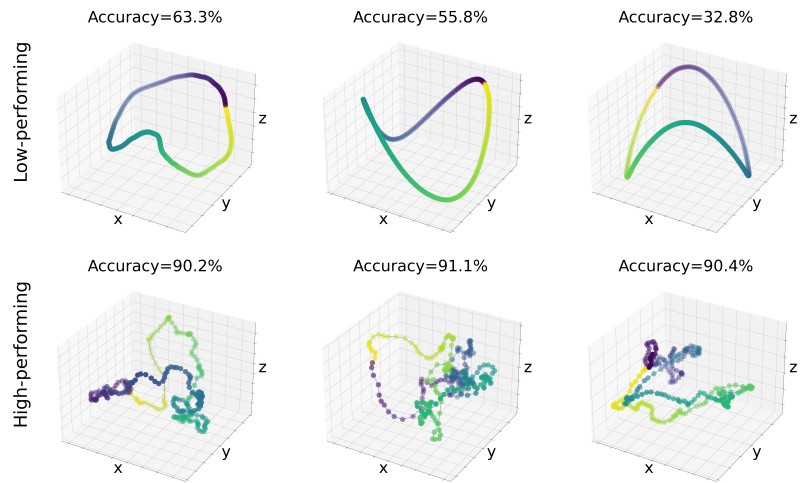

(b) Outputs of six randomly initialised networks induced by a circular input projected on the first three principal components. The accuracies are reported with respect to training on CIFAR-10.

Figure 1: Comparison of convergence, generalisation, and expressivity of low-performing and high-performing networks sampled from NAS-Bench-201 (Dong & Yang, 2020). (a-left): **Generalisation:** The linear independence of feature maps (FMI) is positively correlated to the validation accuracy and hence, generalisation of the network. (a-right): **Convergence:** The networks with high FMI (FMI>25) demonstrate faster convergence during training on CIFAR-10. (b) **Expressivity:** The outputs of the high-performing networks projected on the first three principal components induced by an arbitrary circular input are more curved than those of the low-performing networks. Hence, the low-performing networks are less expressive than the high-performing networks.

2020; White et al., 2021), reinforcement learning methods (Zoph & Le, 2017; Tian et al., 2020), as well as evolutionary algorithms (Real et al., 2019; Chu et al., 2020a) have originally been used to tackle this task. However, they suffer from heavy computational requirements due to the enormous search space. Generative methods (Lukasik et al., 2022; Asthana et al., 2024; An et al., 2024) alleviate this issue by deploying gradient-based learning of the search space through first-order methods. However, the requirement for architectural training data, including the performance of each architecture, poses a serious challenge to the applicability of these methods in practice. Alternatively, zero-shot NAS eliminates the need for such an architecture dataset. This technique leverages the network structure and gradient properties close to initialisation (termed as zero-cost proxies) to predict the performance on a test set prior to its full training (Abdelfattah et al., 2021).

Normally, zero-cost proxies are either task/data agnostic, e.g. FLOPS or parameters, or data-based. Data-agnostic proxies evaluate an architecture's potential without using any input data. They rely solely on the

network's structure to estimate the network performance. In contrast, data-based proxies estimate network performance by passing a small batch of data through the network. While most of the data agnostic proxies (Tanaka et al., 2020; Lin et al., 2021; Li et al., 2021b) fail to be consistent, most data-dependent proxies (Mellor et al., 2021; Abdelfattah et al., 2021) require labels for gradient computation, which are usually not available in real-world scenarios. Thus, these drawbacks motivate the design of a zero-cost proxy independent of labelled data.

Another limitation of most current proxies is their focus on performance attributes that they aim to optimise. Formally, network performance is analysed through three attributes, namely convergence, generalisation, and expressivity (Chen et al., 2023). Convergence refers to how fast a network can converge to a minimum through gradient descent. Generalisation refers to how well a network trained on training data can generalise to unseen test data. Finally, expressivity indicates the complexity of functions that a network can estimate. In addition, the No Free Lunch theorem states that the most optimal network balances these attributes as it is practically impossible to minimise all three simultaneously (Chen et al., 2023). Most zero-cost proxies focus either on convergence and generalisation aspects (Jiang et al., 2023; Li et al., 2023; Yang et al., 2023) or solely on the expressivity characteristic (Lin et al., 2021). However, designing a proxy by balancing all three attributes is an open challenge that is hardly addressed in the literature (Lee & Ham, 2024). This is important because while convergence and generalisation alone are sufficient to somewhat predict the network performance, the architectures searched through NAS methods utilising solely convergence and generalisation are inherently less expressive, either through their limited depth or their choice of operations (as observed in Appendix Section A.8). Hence, the No Free Lunch Theorem states that the best networks are the ones that have balanced convergence, generalisation and expressivity Chen et al. (2023). Therefore, to generate optimal architectures, a zero-cost proxy balancing all three attributes is required.

To address the above shortcomings, we present a zero-cost proxy that (i) operates without the need for labelled data, and (ii) balances convergence, generalisation, and expressivity of networks in one approach. To this end, we first demonstrate using Singular Value Decomposition (SVD) (Klema & Laub, 1980) that the collinearity of feature maps negatively affects the training convergence rate and generalisation capabilities of a Convolutional Neural Network (CNN). In other words, the convergence and generalisation of a CNN are positively correlated to the linear independence of feature maps (FMI), as illustrated in Figure 1a. This makes intuitive sense as a CNN with limited width or repeated identical blocks is structurally constrained in its ability to decorrelate inputs, and hence its feature maps would exhibit high colinearity. Therefore, the given CNN would have poor convergence and generalisation. Next, we employ Riemannian geometry (Lee, 2006) to consider the relationship between the extrinsic curvature of the output and the expressivity of a network (Poole et al., 2016). Specifically, we utilize the fact that the expressivity of a network is related to how curved the output is when an input is fed into the network. This is visually illustrated in Figure 1b. Based on the relationships related to convergence, generalisation, and expressivity, we propose a cost-effective proxy, namely Dextr, which leverages the collinearity of layer feature maps and the extrinsic curvature of the output. As a result, Dextr accurately predicts the network performance on test data using just a single label-free data sample. Our evaluation shows the capabilities of our proxy by outperforming all existing zero-shot NAS methods on three standard tabular benchmarks, specifically NAS-Bench-101 (Ying et al., 2019), NAS-Bench-201 (Dong & Yang, 2020), and TransNAS-Bench-micro (Duan et al., 2021) benchmarks, and showing competitive performance in the NAS-Bench-301 (Zela et al., 2022) benchmark. Additionally, we perform experiments within two search spaces: the DARTS search space (Liu et al., 2019) containing CNNs, and the AutoFormer search space (Chen et al., 2021b) containing Vision Transformers (ViTs) (Dosovitskiy et al., 2021). The architecture discovered through our proxy in these search spaces is then evaluated by training it on ImageNet (Deng et al., 2009) and its test performance is compared with baseline approaches. We demonstrate that the architecture found through our proxy outperforms the previous multi-shot and zero-shot NAS methods on ImageNet classification, all while maintaining minimal search time. We conduct a total of three ablation studies, where we analyse the stability of our approach, compare Dextr with a simple combination of existing proxies, and analyse contribution of individual layers. We additionally perform two experiments on NATS-Bench-SSS (Dong et al., 2021) and MobileNet-v2 (Sandler et al., 2018), provided in the Appendix Section A.4. In summary, our contributions are as follows:

- We present the relationship between the collinearity of feature maps across a convolutional layer in a network and the convergence and generalisation capabilities of the network.

- By incorporating the convergence, generalisation, and expressivity in our proxy, we propose a zero-cost proxy requiring one label-free data sample for its computation.

- Our proposed proxy outperforms the existing methods on three correlation benchmarks, i.e. NAS-Bench-101 (Ying et al., 2019), NAS-Bench-201 (Dong & Yang, 2020), and TransNAS-Bench-101-micro (Duan et al., 2021), as well as on the ImageNet (Deng et al., 2009) NAS task within the DARTS (Liu et al., 2019) search space and the AutoFormer search space (Chen et al., 2021b).

## 2 Related Work

**Multi-shot and Zero-Shot NAS**   Traditional attempts to automate the design process of neural network architectures, such as random search (Li & Talwalkar, 2020), evolutionary approaches (Real et al., 2019; Chu et al., 2020a), reinforcement learning approaches (Zoph & Le, 2017; Tian et al., 2020), and gradient-based approaches (Brock et al., 2018; Yang et al., 2020a; Chen et al., 2021d), are considered rather slow due to the requirement of training iterations at each search step (Liu et al., 2019). More recent generative methods (Lukasik et al., 2022; Asthana et al., 2024; An et al., 2024) addressed this limitation by employing learning-based techniques in the architecture search space. For instance, Lukasik et al. (2022) employed a generator and surrogate predictor to accurately learn and sample from the architectural data distribution. Recently, similar to this direction, An et al. (2024) and Asthana et al. (2024) employed conditioned diffusion models to generate well-performing neural network architectures. However, the expensive training process of the generative models and the requirement of architectural training data limits the applicability of these approaches.

Alternatively, zero-shot NAS accelerates the rather costly architecture search process (Li & Talwalkar, 2020; White et al., 2021; Zoph & Le, 2017; Tian et al., 2020; Real et al., 2019; Chu et al., 2020a) by using network metrics at initialisation, known as zero-cost proxies. The concept of zero-shot NAS was established by Mellor et al. (2021), who introduced the activation overlap between data points as a proxy. Since then, various zero-cost proxies (Lee et al., 2019; Lin et al., 2021; Chen et al., 2021c) have advanced the field, including Snip (Lee et al., 2019), Grasp (Wang et al., 2020), Synflow (Tanaka et al., 2020), Fisher (Turner et al., 2020), and Jacob (Mellor et al., 2021), all proposed by Abdelfattah et al. (2021). Additionally, Li et al. (2023) have introduced ZiCo , which predicts network performance using the mean and standard deviation of parameter gradients at initialisation. Unlike our approach, all these proxies rely on data labels for their computation.

In contrast, some previous works do not require data labels for performance estimation. For instance, NASWOT (Mellor et al., 2021) analyses activation overlaps for network performance estimation. However, it only considers the expressivity of a network and neglects the convergence and generalisation attributes. Similarly, Zen (Lin et al., 2021) considers the expressivity exclusively by utilising the network's Gaussian complexity. Conversely, MeCo (Jiang et al., 2023) omits the need for labels by calculating the minimum eigenvalue of the Pearson Correlation matrix. However, it solely focuses on the convergence and generalisation aspects of the network, while neglecting the expressivity. To consider all three attributes, Chen et al. (2021c) utilised Neural Tangent Kernels (NTK) and the Number of Linear Regions (NLR) for efficient architecture search. More recently, AZ-NAS (Lee & Ham, 2024) considers multiple network attributes like trainability, expressivity, progressivity, and complexity. However, they both require labels and backpropagation for their calculation. Following a more efficient direction, we propose a zero-cost proxy that further improves the performance without requiring labels while considering all three performance attributes, i.e. convergence, generalisation, and expressivity.

**Performance Attributes**   Several works have studied the theory concerning convergence, generalisation and expressivity of the neural networks (Neal, 1996; Williams, 1996; Du et al., 2019a; Poole et al., 2016). For instance, Du et al. (2019a;b) provided insights into the convergence and generalisation of overparameterized networks through gram matrix analysis. Building upon this idea, we establish a connection between the collinearity of feature maps and the convergence and generalisation capabilities of CNNs. Additionally, Poole

et al. (2016) investigated the expressivity of neural networks using concepts from Riemannian geometry (Lee, 2006) and Mean Field Theory (Weiss, Pierre, 1907). We leverage these concepts to incorporate expressivity into our proxy. Finally, Chen et al. (2023) demonstrated the 'No Free Lunch' behaviour, indicating that with a fixed number of parameters, no single architecture can simultaneously optimise all performance attributes. This is because convergence and generalisation properties favour wide and shallow networks while expressivity favours deep and narrow network topologies. Hence, our proxy aims to balance these attributes rather than optimising all three simultaneously.

## 3 Dextr

### 3.1 Problem Formulation

Consider the convergence, generalisation, and expressivity of an overparameterised neural network. In this context, convergence refers to the rate at which the loss of this network reaches a minimum, generalisation refers to the performance of the network on unseen test data post-training, and expressivity refers to the ability of the network to model complex functions.

We aim to design a zero-cost proxy that considers all three attributes – convergence, generalisation, and expressivity of a network without the requirement for labelled data. To this end, based on the background concepts detailed in Section 3.2, we establish the relationship between the collinearity of feature maps of a specific convolutional layer in the CNN with the convergence and generalisation properties of this network, by employing Singular Value Decomposition. To address expressivity, we build upon the findings of Poole et al. (2016) (Section 3.2.2) for designing our proxy. Next, we apply our proposed proxy to Vision Transformers (ViTs) (Section 3.3.4). Finally, we utilise the proposed proxy to search for optimal neural architectures in a training-free setting.

### 3.2 Background Theory

We consider, for simplicity, the two-layer fully-connected neural network $f(\mathbf{W}, \mathbf{a}, \mathbf{x}_i)$ with ReLU activation function, as Du et al. (2019b), such that

$$f(\mathbf{W}, \mathbf{a}, \mathbf{x}_i) = \frac{1}{\sqrt{m}} \sum_{r=1}^{m} a_r \sigma(\mathbf{w}_r^T \mathbf{x}_i), \tag{1}$$

where $\mathbf{x}_i$ is the input to $f$ taken from the training set $\mathcal{D} = \{(\mathbf{x}_i, y_i)\}_{i=1}^{n}$, $y_i$ is the ground-truth for $\mathbf{x}_i$, $\mathbf{w} \in \mathbb{R}^d$ and $\mathbf{W} \in \mathbb{R}^{d \times m}$ are the weight vector and matrix for the first layer respectively, $m$ is the number of hidden nodes, $\sigma$ is the ReLU activation function, $a_r \in \mathbb{R}$ is the output weight, and $\mathbf{a} \in \mathbb{R}^{m \times 1}$ is the output weight vector. We consider the simple case where the training objective of this network is to minimise the following Mean Squared Error (MSE) loss (Du et al., 2019b):

$$L(\mathbf{W}, \mathbf{a}) = \sum_{i=1}^{n} \frac{1}{2} (f(\mathbf{W}, \mathbf{a}, \mathbf{x}_i) - y_i)^2. \tag{2}$$

First, we present the background theory regarding convergence, generalisation, and expressivity of $f$. Then, we demonstrate the transferability of these foundational concepts to CNNs.

#### 3.2.1 Convergence and generalisation: Minimum eigenvalue of gram matrix

We start by examining the convergence and generalisation of $f$. Following Du et al. (2019b), consider a continuous training scenario where $f$ is trained through gradient descent with infinitely small step-size $t$, and the output layer of $f$ is fixed. The prediction of this network on a given input $\mathbf{x}_i$ at training time step $t$ is denoted as $u_i(t) = f(\mathbf{W}(t), \mathbf{a}, \mathbf{x}_i)$. Then, the gram matrices generated by the ReLU activation function on the training set $\mathcal{D}$ at a given time $t$ and at initialisation, i.e. $t = 0$, respectively are defined as

$$[\mathbf{H}(t)]_{ij} = \frac{1}{m} \sum_{r=1}^{m} \mathbf{x}_i^T \mathbf{x}_j \mathbb{I}\{\mathbf{w}_r^T(t)\mathbf{x}_i \geq 0, \mathbf{w}_r^T(t)\mathbf{x}_j \geq 0\},$$

$$\mathbf{H}_{ij}^{\infty} = \mathbb{E}_{\mathbf{w} \sim \mathcal{N}(\mathbf{0}, \mathbf{I})}[\mathbf{x}_i^T \mathbf{x}_j \mathbb{I}\{\mathbf{w}^T \mathbf{x}_i \geq 0, \mathbf{w}^T \mathbf{x}_j \geq 0\}],$$

where $r \in [m]$, $\mathcal{N}\{\cdot\}$ represents the Gaussian distribution, $\mathbb{I}$ represents the indicator function and $\mathbf{H}(t) \in \mathbb{R}^{n \times n}, \mathbf{H}^{\infty} \in \mathbb{R}^{n \times n}$ are the respective gram matrices. Next, we consider the following theorem for network convergence (Du et al., 2019b).

**Theorem 1** *Let the input be $i \in [n]$, $c_{low} < \|\mathbf{x}_i\|_2 < c_{high}$ and $\|y_i\| < C$, where $c_{high}$, $c_{low}$, and $C$ are constants. If the number of nodes $m$ is set to $\Omega(\frac{n^6}{\lambda_{min}(\mathbf{H}^{\infty})^4 \delta^6})$, where $\delta$ is the probability of failure, $\Omega$ denotes the lower bound, and $\lambda_{min}(\mathbf{H}^{\infty})$ denotes the scaled minimum eigenvalue of $\mathbf{H}^{\infty}$, and we i.i.d. initialise $\mathbf{w}_r \sim \mathcal{N}(\mathbf{0}, \mathbf{I})$ and $a_r \sim \mathcal{U}\{[-1, 1]\}$, where $\mathcal{U}\{\cdot\}$ represents the uniform distribution, then with at least $1 - \delta$ probability, the following relationship holds:*

$$\|\mathbf{u}_i(t) - \mathbf{y}_i\|_2^2 \leq \exp(-\lambda_{min}(\mathbf{H}^{\infty})t)\|\mathbf{u}_i(0) - \mathbf{y}_i\|_2^2. \tag{3}$$

Theorem 1 establishes that the loss incurred by $f$ at time-step $t$, defined as $\|\mathbf{u}(t) - \mathbf{y}\|_2^2$, has an upper bound controlled by $\lambda_{min}(\mathbf{H}^{\infty})$. If we take the training loss at initialisation out of consideration, then higher the term $\lambda_{min}(\mathbf{H}^{\infty})$ would be, lower the term $\exp(-\lambda_{min}(\mathbf{H}^{\infty})t)$ would be, and lower the upper bound of the training loss at time t, i.e. $\|\mathbf{u}_i(t) - \mathbf{y}_i\|_2^2$ would be, which shows better convergence. In conclusion, the term $\lambda_{min}(\mathbf{H}^{\infty})$ at a given time $t$ controls the upper bound of the training loss at $t$ and hence, governs the convergence rate of $f$. The proof of this theorem is provided in the work by Du et al. (2019b). Theorem 1 can be extended to the case of deep neural networks, as described in the Appendix Section A.2. Next, we explore network generalisation based on the following theorem from Cao & Gu (2019) and Zhu et al. (2022).

**Theorem 2** *Let the loss function of $f$ with $m \to \infty$ evaluated on the test set be denoted as $L(\mathbf{W})$. Let the ground truth $\mathbf{y} = (y_1, ..., y_N)^T$, where $N$ is the total number of samples and $\gamma$ represents the step size of stochastic gradient descent (SGD), determined by $\gamma = kC_1 \sqrt{\mathbf{y}^T (\mathbf{H}^{\infty})^{-1} \mathbf{y}}/(m\sqrt{N})$, where $k$ is a sufficiently small absolute constant (Zhu et al., 2022). If Theorem 1 holds, then for any $\delta \in (0, e^{-1}]$, there exists a value $m^*$ which depends on $\delta, N$, and $\lambda_{min}(\mathbf{H}^{\infty})$ such that if $m \geq m^*$, with a probability of at least $1 - \delta$, the following inequality holds:*

$$\mathbb{E}[L(\mathbf{W})] \leq \mathcal{O}\left(C_2 \sqrt{\frac{\mathbf{y}^T \mathbf{y}}{\lambda_{min}(\mathbf{H}^{\infty})N}}\right) + \mathcal{O}\left(\sqrt{\frac{\log(1/\delta)}{N}}\right), \tag{4}$$

where $C_1$ and $C_2$ are constants.

We observe that the upper bound of the expected test loss, $\mathbb{E}[L(\mathbf{W})]$, is governed by two key terms. In particular, the smaller the term $\mathcal{O}\left(C_2 \sqrt{\frac{\mathbf{y}^T \mathbf{y}}{\lambda_{min}(\mathbf{H}^{\infty})N}}\right)$ is, the tighter the upper bound on the test loss. Consequently, a larger value of $\lambda_{min}(\mathbf{H}^{\infty})$ leads to a smaller test loss, indicating better generalisation. Therefore, $\lambda_{min}(\mathbf{H}^{\infty})$ positively correlates with the generalisation capabilities of $f$. The proof of this theorem is detailed in the work by Jiang et al. (2023).

**Remark 1** *Based on Theorems 1 and 2 and findings from Du et al. (2019b), while $\mathbf{H}(t)$ changes through the progression of $t$ from 0 to $\infty$, it still stays close if $m \to \infty$. Moreover, for any two non-parallel inputs $\mathbf{x}_i$ and $\mathbf{x}_j$ (i.e. $\mathbf{x}_i \nparallel \mathbf{x}_j$), the minimum eigenvalue of the gram matrix $\mathbf{H}^{\infty}$ at initialisation, i.e. $\lambda_{min}(\mathbf{H}^{\infty})$ is strictly positive, i.e. $\lambda_{min}(\mathbf{H}^{\infty}) > 0$, and is positively correlated to the convergence rate and generalisation capabilities of $f$.*

The relationship between $\mathbf{H}(t)$ and $\mathbf{H}^{\infty}$ in Remark 1 also implies that the inputs $\mathbf{x}_i(t)$ and $\mathbf{x}_j(t)$ at any training time step $t$ can be used to estimate the convergence rate and generalisation of $f$. This motivates us to design our proxy close to the start of the training, i.e. when $t$ is close to 0.

### 3.2.2 Expressivity: Extrinsic curvature of the output

Next, we consider the expressivity characteristics of $f$ using the characterisation through extrinsic curvature of the output, given by Poole et al. (2016). We utilise this characterisation because the extrinsic curvature

of the output captures how sensitive the network's output is to small changes in its input. This property is directly connected to the expressivity as an expressive model can adapt finely to differences between input samples, which is reflected in high local curvature. To examine the expressivity of $f$, we leverage principles from Riemannian geometry (Lee, 2006). In this geometry, a manifold represents a topological space that resembles Euclidean space locally near each point but may have an intricate global structure. Following Poole et al. (2016), we define $\mathbf{g}(\theta)$ as a 1-dimensional manifold within the input space, where $\theta$ serves as an intrinsic scalar coordinate on this manifold. We consider $\mathbf{g}(\theta)$ to be an arbitrary circular input i.e. $\mathbf{g}(\theta) = \sqrt{N_1 q}[\mathbf{o}^0 cos(\theta) + \mathbf{o}^1 sin(\theta)]$, where $\theta \in [0, 2\pi)$, and $\mathbf{o}^0$, $\mathbf{o}^1$ form an orthonormal basis for a 2-dimensional subspace of $\mathbb{R}^{N_1}$. The layer propagation of $f$ transforms this input manifold into a new manifold $\mathbf{h}^l(\theta) = \mathbf{h}^l(\mathbf{g}(\theta))$, where $l$ corresponds to the layers of $f$. Each point $\theta$ in the manifold $\mathbf{h}^l(\theta)$ induces a velocity vector (equal to its tangent), represented by $\mathbf{v}^l(\theta) = \partial_\theta \mathbf{h}^l(\theta)$, and an acceleration vector $\mathbf{a}^l(\theta) = \partial_\theta \mathbf{v}^l(\theta)$. Formally, the extrinsic curvature of the output, induced by the arbitrary circular input $\mathbf{g}(\theta)$ parameterised by $\theta$ is defined as in Poole et al. (2016):

$$\kappa(\theta) = (\mathbf{v} \cdot \mathbf{v})^{-3/2}\sqrt{(\mathbf{v} \cdot \mathbf{v})(\mathbf{a} \cdot \mathbf{a}) - (\mathbf{v} \cdot \mathbf{a})^2}. \tag{5}$$

**Remark 2** *The extrinsic curvature of the output, i.e. $\kappa(\theta) \in \mathbb{R}$ measures the complexity of the output curve, which indicates the functional complexity of $f$. Thus, $\kappa(\theta)$ is positively correlated to the expressivity characteristic of the network $f$.*

Intuitively, a more expressive network induces a more tangled output manifold after propagation through the network, as observed in Figure 1b. Thus it has a higher $\kappa(\theta)$. The extrinsic curvature $\kappa(\theta)$ remains invariant irrespective of the particular parameterisation $\theta$ and does not require any data for its computation. Moreover, this characterisation of expressivity is applicable to networks with random weights (Poole et al., 2016), i.e. at time-step $t = 0$. The data-independent property of $\kappa(\theta)$, as well as its ability to express the functional complexity of $f$ at network initialisation makes it suitable to incorporate $\kappa(\theta)$ in our proxy.

### 3.2.3 Approximation to convolutional layer

Consider the multi-channel convolutional layer $l^{cnn}$ of the CNN $f_{cnn}$. The number of channels in layer $l^{cnn}$ is denoted as $C$. The vector-represented output feature maps at time step $t$ are denoted as $\mathbf{x}_c^\phi(t)$, where $c \in \{1, 2, \cdots, C\}$. We define $\mathbf{X}^\phi(t)$ as a matrix containing all the vector-represented feature maps, such that $\mathbf{x}_c^\phi(t) \in \mathbf{X}^\phi(t)$. The network $f_{cnn}$ is trained on the training set $\mathcal{D} = \{(\mathbf{x}_i, y_i)\}_{i=1}^n$. The output of $f_{cnn}$ at time step $t$ is denoted as $\mathbf{u}_i^\phi(t)$ and the loss incurred by $f_{cnn}$ at time step $t$ is defined as $\|\mathbf{u}_i^\phi(t) - \mathbf{y}_i\|_2^2$.

**Remark 3** *Through the approximation of the multi-sample fully-connected layer of $f$ to a multi-channel convolutional layer $l^{cnn}$ of the network $f_{cnn}$ under some constraints presented by Jiang et al. (2023), we know that each data sample $\mathbf{x}_i$ can be regarded as a vector-represented channel $\mathbf{x}_c^\phi$ of the convolutional layer $l^{cnn}$, where $c \in \{1, C\}$. Thus, the concepts regarding $f$ presented in Section 3.2.1 and 3.2.2 hold for $f_{cnn}$ by replacing the input data sample $\mathbf{x}_i$ with the feature maps $\mathbf{x}_c^\phi$.*

Hence, the gram matrix $\mathbf{H}(t)$ for multiple input samples in the fully-connected layer of $f$ can be approximated as $\mathbf{H}^\phi(t)$ for multiple channels in the convolutional layer $l^{cnn}$, where $\mathbf{H}^\phi(t)$ is the gram matrix associated to the feature maps at time step $t$. A detailed explanation on approximation to convolutional layer is provided in the Appendix Section A.3.

### 3.3 Designing Dextr

We now design our zero-cost proxy, focused on CNNs (Sec. 3.2.3), by utilising the remarks on the convergence and generalisation (Sec. 3.2.1), as well as expressivity (Sec. 3.2.2). In particular, we exploit the collinearity (Sec. 3.3.1) among all channels of the layers in the CNN using SVD (Sec. 3.3.2) to formulate our proxy, namely Dextr.

### 3.3.1 Multi-collinearity in feature maps

According to Remark 1, we observe that firstly $\lambda_{min}(\mathbf{H}^\infty)$ correlates positively with the training convergence rate and the generalisation capabilities of $f$. Furthermore, for any two non-parallel inputs $\mathbf{x}_i$ and $\mathbf{x}_j$, $\mathbf{H}(t)$ remains close to $\mathbf{H}^\infty$ at each training time step $t$. Therefore, when $t$ is close to 0, $\lambda_{min}(\mathbf{H}(t))$ can similarly measure the convergence and generalisation of $f$. In Remark 3, we recall the approximation of a fully-connected layer to the convolutional layer (Jiang et al., 2023) by replacing the input sample $\mathbf{x}_i(t)$ with the vector-represented channel $\mathbf{x}_c^\phi(t)$ and $\mathbf{H}(t)$ with $\mathbf{H}^\phi(t)$. Therefore, we infer that $\lambda_{min}(\mathbf{H}^\phi(t))$ *of any convolutional layer is positively correlated with the convergence and generalisation of a CNN.*

Additionally, the eigenvalues of $\mathbf{H}^\phi(t)$ represent the variance along the orthogonal eigenvectors. These eigenvectors capture the orthogonal directions of the variability of the data (Shawe-Taylor et al., 2005). Consequently, $\lambda_{min}(\mathbf{H}^\phi(t))$ *shows the uniqueness of information in the most linearly dependent channel or feature map in* $\mathbf{X}^\phi(t)$. Lower values of $\lambda_{min}(\mathbf{H}^\phi(t))$ indicate that at least one channel in $\mathbf{X}^\phi(t)$ is highly redundant and could be represented as a linear combination of other channels.

Instead of focusing solely on the linear dependence or collinearity of the most linearly dependent channel in $\mathbf{X}^\phi(t)$, we focus on the collinearity among all channels jointly. Thus, we assume that the convergence and generalisation of $f_{cnn}$ are negatively correlated to the multi-collinearity of channels in $\mathbf{X}^\phi(t)$. This assumption is proved later in Theorems 3 and 4.

### 3.3.2 Singular Value Decomposition for multi-collinearity

To estimate the multi-collinearity in $\mathbf{X}^\phi(t)$, we deploy SVD due to its effectiveness and numerical stability, especially when considering large ill-conditioned matrices (Klema & Laub, 1980). We know that the singular values $\sigma_k$ of $\mathbf{X}^\phi(t)$ are equal to the square root of the eigenvalues $\lambda_k$ of the gram matrix $\mathbf{H}^\phi(t)$ at any time-step $t$, i.e. $\sigma_k = \sqrt{\lambda_k(\mathbf{H}^\phi(t))}$, where $k$ indexes the orthogonal eigenvectors.

**Remark 4** *The condition number of* $\mathbf{X}^\phi(t)$, *defined as the ratio of the maximum singular value to the minimum singular value, i.e.* $c(\mathbf{X}^\phi(t)) = \sigma_{max}(\mathbf{X}^\phi(t))/\sigma_{min}(\mathbf{X}^\phi(t)) \in \mathbb{R}$, *calculated through SVD, depicts the ill-conditioning or collinearity of vectors in* $\mathbf{X}^\phi(t)$ *(Demmel, 1987).*

Thus, we utilise the condition number $c(\mathbf{X}^\phi(t))$ to establish the relationship between the collinearity of channels in $\mathbf{X}^\phi(t)$ and the convergence and generalisation properties of the CNN $f_{cnn}$. Hence, we present the following lemma and theorems concerning convergence and generalisation through adaptation of the theorems presented in Section 3.2.1.

**Lemma 1** *Consider a multi-channel convolutional layer with the number of channels as $C$, then the maximum singular value of* $\mathbf{X}^\phi(t)$ *is greater than or equal to 1, i.e.* $\sigma_{max}(\mathbf{X}^\phi(t)) \geq 1$. *The proof and the experimental validation is provided in the Appendix Sections A.1.3 and A.1.4 respectively.*

**Theorem 3** *Consider a multi-channel convolutional neural network $f_{cnn}$ that is approximated by the network $f$. Let the input to $f$ be $i \in [n]$ and $c_{low} < ||\mathbf{x}_i||_2 < c_{high}$, $|y_i| < C$ for some constant $c_{low}$, $c_{high}$, and $C$. Next, consider the Lemma 1 to be satisfied, and that the number of nodes $m$ for $f$ is selected according to $\Omega(\frac{n^6}{\lambda_{min}(\mathbf{H}^\infty)^4 \delta^6})$, where $\Omega$ represents the lower bound. Then, if we perform an i.i.d. initialisation of $\mathbf{w}_r \sim \mathcal{N}(\mathbf{0}, \mathbf{I})$ and let $a_r \sim \mathcal{U}\{[-1,1]\}$ for $r \in [m]$, where $\mathcal{U}$ denotes a uniform distribution, then with a probability of at least $1 - \delta$, the following inequality holds for $f_{cnn}$:*

$$\|\mathbf{u}_i^\phi(t) - \mathbf{y}_i\|_2^2 \leq \exp(-t/c(\mathbf{X}_i^\phi(t=0))^2)\|\mathbf{u}_i^\phi(0) - \mathbf{y}_i\|_2^2, \tag{6}$$

*where* $\mathbf{X}_i^\phi(t=0)$ *is the scaled feature matrix at initialisation, i.e. $t = 0$.*

We observe from Theorem 3 that $c(\mathbf{X}^\phi(t = 0))^2$ *negatively correlates with the convergence rate.* Thus, through Remark 4 and Theorem 3, we deduce that the convergence rate worsens with the increase in multi-collinearity of $\mathbf{X}^\phi(t=0)$. The proof for Theorem 3 is available in the Appendix Section A.1.

**Theorem 4** *For the test loss function $L^\phi(\mathbf{W})$ of the network $f_{cnn}$, let the SGD step-size $\gamma$ be determined by $\gamma = kC_1\sqrt{\mathbf{y}^T(\mathbf{H}^\infty)^{-1}\mathbf{y}}/(m\sqrt{N})$, where $k$ is a small absolute constant. Let the ground truth $\mathbf{y} = (y_1, ..., y_N)^T$, where $N$ is the number of samples. If Theorem 3 holds, then for any $\delta \in (0, e^{-1}]$, there exists a value $m^*$ which depends on $\delta, N$, and $c(\mathbf{X}^\phi(t=0))$ such that if $m \geq m^*$, then with a probability of at least $1 - \delta$, the following inequality holds:*

$$\mathbb{E}[L^\phi(\mathbf{W})] \leq \mathcal{O}\left(C_2 c(\mathbf{X}^\phi(t=0))\sqrt{\frac{\mathbf{y}^T\mathbf{y}}{N}}\right) + \mathcal{O}\left(\sqrt{\frac{\log(1/\delta)}{N}}\right), \tag{7}$$

*where $C_1$ and $C_2$ are constants.*

The proof for this theorem is presented in the Appendix Section A.1. We observe from Theorem 3 that the upper bound of the test loss of $f_{cnn}$ is directly proportional to $c(\mathbf{X}^\phi(t=0))$. Hence, $c(\mathbf{X}^\phi(t=0))$ is inversely proportional to the generalisation capabilities of $f_{cnn}$.

We now see from Theorem 3 and 4 that *the convergence and generalisation of $f_{cnn}$ are negatively correlated to the multi-collinearity of feature maps of a given layer at initialisation*, which is also observed in Figure 1a. Next, we formulate Dextr through the proposed theorems.

### 3.3.3 Formulating Dextr

Through the Theorem 3 and 4, we note that $c(\mathbf{X}^\phi(t=0))$ correlates negatively with the convergence and generalisation of $f_{cnn}$. Through Remark 1, we know that $\mathbf{H}(t)$ stays close to $\mathbf{H}^\infty$ (when $t=0$) if $m \to \infty$. Hence, we infer that $\lambda_{min}(\mathbf{H}^\phi(t))$ approximates the value of $\lambda_{min}(\mathbf{H}^\phi(t=0))$. Thus, through the relation $\sigma_k = \sqrt{\lambda_k(\mathbf{H}^\phi(t))}$, we know that the condition number $c(\mathbf{X}^\phi(t))$ also approximates the value $c(\mathbf{X}^\phi(t=0))$. Hence, *the inverse of $c(\mathbf{X}^\phi(t))$ correlates positively with the convergence and generalisation*. Subsequently, through Remark 2, we observe that *the extrinsic curvature of the output, i.e. $\kappa(\theta)$ correlates positively with the expressivity characteristics of $f_{cnn}$*. To balance convergence, generalisation, and expressivity characterisations, we propose the following zero-cost proxy using the simplified harmonic mean of the logarithms of the extrinsic curvature $\kappa(\theta)$ and the sum of the inverse of the feature condition number $c(\mathbf{X}^\phi(t))$:

$$Dextr = \frac{\log\left(1 + \sum_{l=1}^{L}\frac{1}{c_l(\mathbf{X}^\phi)}\right) \cdot \log\left(1 + \kappa(\theta)\right)}{\log\left(1 + \sum_{l=1}^{L}\frac{1}{c_l(\mathbf{X}^\phi)}\right) + \log\left(1 + \kappa(\theta)\right)} \tag{8}$$

where L is the number of layers, $c_l(\mathbf{X}^\phi)$ is the condition number corresponding to the feature maps of layer $l$ and $\kappa(\theta)$ is the extrinsic curvature of the output given an arbitrary circular input $\theta$. As observed from Eq. 8, the proposed proxy is defined as the half of the harmonic mean between two terms: $\log\left(1 + \sum_{l=1}^{L}\frac{1}{c_l(\mathbf{X}^\phi)}\right)$ and $\log\left(1 + \kappa(\theta)\right)$. The logarithmic transformations in Eq. 8 are applied to reduce the effect of the scale of these two terms and to avoid numerical instabilities, while the addition of 1 ensures that both terms remain positive. Note that we omit the training time step $t$ in our proposed proxy as Dextr is invariant to $t$, meaning theoretically, feature maps $\mathbf{X}^\phi$ and curvature $\kappa(\theta)$ at any point in training can be used to calculate Dextr.[*]

### 3.3.4 Application to Vision Transformers

To demonstrate the application of Dextr to Vision Transformers (ViTs) (Dosovitskiy et al., 2021), we rely on the relation of CNNs with ViTs given by Li et al. (2021a). Notably, Li et al. (2021a) prove that a multi-head self-attention (MSA) layer in ViTs with relative positional encoding can exactly mimic a convolution operation if the number of heads is carefully set with a sufficient number, mainly due to the similarities between the attention mechanism and convolutions. This supports the applicability of our convolutional layer analysis (Section 3.2.3) to ViTs. Next, we consider the Gaussian Error Linear Units (GeLU) activation function (Hendrycks & Gimpel, 2016) used in ViTs. Since GeLU is regarded as a smooth approximation

---

[*]We provide the procedure to calculate Dextr and the experimental settings in the Appendix Sections A.6 and A.7 respectively.

of ReLU (Hendrycks & Gimpel, 2016), the behaviour of these functions is comparable. Thus, the concepts discussed earlier in relation to ReLUs can be directly applied to GeLUs as well.

Lastly, we acknowledge that in some cases, strict equivalence of ViTs and CNNs is not met, specifically, in the cases where relative positional encodings are not always used, or where the number of attention heads may not be large enough to achieve the convolutional limit. However, we argue that the reason why Dextr is generalisable on ViTs is because a strict equivalence is not necessary for Dextr to be effective. As long as the output of the transformer layers exhibit somewhat sufficient structure similar to CNNs, the SVD analysis is meaningful. Importantly, our theoretical derivation interprets inputs or features as collections of vectors, on which SVD analysis is performed. Notably, the central assumption of our theory is not convolution, but rather the existence of a meaningful Gram matrix formed from activations. Since ViT block outputs can be reshaped or projected into such vector sets, our analysis of convergence and generalisation via Gram eigenvalues, and hence SVD, extends naturally.

## 4 Experiments

Our evaluation is performed through two standard protocols (Li et al., 2023; Jiang et al., 2023; Sun et al., 2023), namely the correlation experiments and NAS experiments. The experiments include a total of six benchmarks involving CNN and ViT architectures. We evaluate our method on a variety of supervised and self-supervised tasks, including image classification (Ying et al., 2019; Dong & Yang, 2020; Zela et al., 2022), object classification (Duan et al., 2021), scene classification (Duan et al., 2021), and jigsaw classification (Duan et al., 2021).

### 4.1 Correlation experiments

**Experimental Protocol**   We first consider the correlation experiments. To this end, we evaluate our proposed zero-cost proxy on four standard benchmarks: NAS-Bench-101 (Ying et al., 2019), NAS-Bench-201 (Dong & Yang, 2020), NAS-Bench-301 (Zela et al., 2022), and TransNAS-Bench-101-micro (Duan et al., 2021) through the NAS-Bench-Suite-Zero framework (Krishnakumar et al., 2022) for an unbiased comparison (Peng et al., 2024). Additionaly, we provide correlation experiments on NATS-Bench-SSS (Dong et al., 2021) in the Appendix Section A.5. To evaluate our approach, we utilise the standard evaluation protocol (Peng et al., 2024; Jiang et al., 2023; Li et al., 2023), calculating the Spearman rank correlation coefficient (Zar, 2005) between the proxy value and the architecture test accuracy. We consider 1000 randomly sampled architectures from NAS-Bench-Suite-Zero for our analysis. We perform 3 runs of each correlation experiment and compare the mean proxy-accuracy correlation of Dextr with several zero-cost proxies, following the same evaluation protocol, namely Fisher (Turner et al., 2020), Snip (Lee et al., 2019), Grasp (Wang et al., 2020), Synflow (Tanaka et al., 2020), Grad_Norm (Abdelfattah et al., 2021), MeCo (Jiang et al., 2023), SWAP-NAS (Peng et al., 2024), and the number of parameters #params.

**Discussion of Results**   Table 1 presents the comparison of Spearman rank correlation coefficient $\rho$ across different zero-cost proxies. Our method consistently surpasses previous baselines across the NAS-Bench-101 and NAS-Bench-201, and in two out of three tasks of TransNAS-Bench-101-micro benchmark. Specifically, the results in TransNAS-Bench-101-micro benchmark show that Dextr is robust to task variations using the same search space. Moreover, we can observe that our method performs better in scene classification task compared to object classification and jigsaw classification tasks. This is because the scene classification task is an easier task that benefits more from global image cues and spatial hierarchies (Duan et al., 2021), which are more aligned with the characteristics that zero-cost proxies like our method can capture. In particular, our method outperforms other proxies in Scene Classification because it captures the expressive capacity of a network to model global spatial patterns and hierarchical features, both of which are critical for scene understanding.

Remarkably, in the case of NAS-Bench-101, our approach shows an improvement of 8% over the previous best approach. In the NAS-Bench-301 benchmark, our method performs competitively with a correlation of 44%. However, compared to other benchmarks, our method performs relatively worse in this benchmark since NAS-Bench-301 is known to be a particularly challenging benchmark due to high architectural diversity

Table 1: Comparison of Spearman rank correlation coefficient for various zero-cost proxies on four tasks of TransNASBench101-micro (TNB), along with NAS-Bench-101 (NB101), NAS-Bench-201 (NB201), and NAS-Bench-301 (NB301). We report the mean and standard deviation of our results across 3 runs. The best approach is indicated as **bold** and the second-best approach is indicated as underlined. All numbers except ours and MeCo (Jiang et al., 2023) are taken from (Peng et al., 2024).

| Approach | TNB-Object | TNB-Scene | TNB-Jigsaw | NB101 | NB201-cf10 | NB201-cf100 | NB201-ImNet120 | NB301 |
|---|---|---|---|---|---|---|---|---|
| Grasp | -0.22 | -0.43 | -0.12 | 0.27 | 0.39 | 0.46 | 0.45 | 0.34 |
| Fisher | 0.44 | -0.13 | 0.30 | -0.28 | 0.40 | 0.46 | 0.42 | -0.28 |
| Grad_Norm | 0.39 | -0.33 | 0.36 | -0.24 | 0.42 | 0.49 | 0.47 | -0.04 |
| Snip | 0.45 | -0.14 | 0.41 | -0.19 | 0.43 | 0.49 | 0.48 | -0.05 |
| Synflow | 0.48 | 0.27 | 0.47 | 0.31 | 0.74 | 0.76 | 0.75 | 0.18 |
| #params | 0.45 | 0.32 | 0.44 | 0.38 | 0.72 | 0.73 | 0.69 | 0.46 |
| ZiCo | 0.50 | 0.70 | 0.52 | $0.41^{\dagger}$ | 0.76 | 0.79 | 0.77 | $\mathbf{0.49}^{\dagger}$ |
| MeCo | **0.58** | 0.62 | 0.45 | $\underline{0.57}^{\dagger}$ | 0.89 | 0.88 | 0.85 | $0.43^{\dagger}$ |
| Swap_reg | 0.48 | 0.67 | 0.47 | $0.49^{\dagger}$ | 0.88 | 0.90 | **0.87** | $\mathbf{0.49}^{\dagger}$ |
| **Dextr (ours)** | 0.53±0.007 | **0.75±0.003** | **0.55±0.003** | **0.65±0.010** | **0.90±0.006** | **0.91±0.004** | **0.87±0.006** | 0.44±0.007 |

Table 2: Quantitative comparison of networks chosen by the NAS methods on ImageNet (Deng et al., 2009) within the DARTS (Liu et al., 2019) search space in terms of the top-1 and top-5 error rate, along with the number of parameters (Params), Method Type (One Shot, Zero-Shot, or Predictor) and search cost in GPU days. The best approach is indicated as **bold** and the second-best approach is indicated as underlined.

| Method | Top-1/Top-5 ↓ Error | Params (M) | GPU Days ↓ | Type |
|---|---|---|---|---|
| DARTS (2nd) (Liu et al., 2019) | 26.7 / 8.7 | 4.7 | 4.0 | OS |
| SNAS (Xie et al., 2018) | 27.3/9.2 | 4.3 | 1.5 | OS |
| FairDARTS (Chu et al., 2020b) | 24.4/7.4 | 5.3 | 3.8 | OS |
| PC-DARTS (Xu et al., 2019) | 25.1/ 7.8 | 5.3 | 0.1 | OS |
| P-DARTS (Chen et al., 2019) | 24.4 / 7.4 | 4.9 | 0.3 | OS |
| $L^2$-NAS (Mills et al., 2021) | 24.6/7.5 | 5.4 | 0.1 | OS |
| $\beta$-DARTS (Ye et al., 2022) | **23.9 / 7.0** | 5.5 | 0.4 | OS |
| PRE-NAS (Peng et al., 2022) | 24.0/7.8 | 6.2 | 0.6 | PR |
| TE-NAS (Chen et al., 2021c) | 26.2 / 8.3 | 6.3 | 0.05 | ZS |
| ZiCo $^{\dagger}$ (Li et al., 2023) | 24.9/7.6 | 7.7 | 0.02 | ZS |
| MeCo $^{\dagger}$ (Jiang et al., 2023) | 25.1/7.7 | 7.8 | 0.04 | ZS |
| **Dextr (ours)** | **24.6/7.4** | 6.6 | 0.07 | ZS |

(Zela et al., 2022). Notably, our approach demonstrates superior performance in both supervised tasks (scene classification) and self-supervised tasks (jigsaw classification), highlighting its versatility and effectiveness across a range of computer vision tasks. Lastly, as evident from the low standard deviation values, we observe that the results of our approach are stable across different runs.

## 4.2 NAS experiments

**Experimental Protocol** Our NAS experiments involve searching architectures for the ImageNet (Deng et al., 2009) classification task and comprise two search spaces- AutoFormer (Chen et al., 2021b) search space containing Vision Transformer (ViT) architectures and DARTS search space containing CNNs (Liu et al., 2019). We conduct an additional experiment on MobileNet-v2 space (Sandler et al., 2018), available in the Appendix Section A.5.2.

Our DARTS evaluation follows the standard protocol (Liu et al., 2019; Chen et al., 2021c; Peng et al., 2022) of searching for the optimal architecture on CIFAR-10 and training the found architecture on ImageNet (Deng et al., 2009). Furthermore, the search for the best architecture for all zero-cost proxies is based on the operation scoring strategy through the Zero-Cost-PT algorithm (Xiang et al., 2023). We perform the search and training procedures and report the top-1 and top-5 error of the architecture found by Dextr, along with the number of parameters and search cost in GPU days, and compare with various baseline one-shot and zero-shot NAS approaches, including DARTS (Liu et al., 2019), SNAS (Xie et al., 2018), FairDARTS (Chu

Table 3: Quantitative comparison for the AutoFormer (Chen et al., 2021b) search space in the Tiny and Small setting with parameter constraints 6M and 24M respectively. We report the top-1 error, the number of parameters (Params) and floating point operations (FLOPs) of selected networks on ImageNet (Deng et al., 2009), along with the search cost in GPU days. The best approach is indicated as **bold** and the second-best approach is indicated as underlined.

| Method | Top-1 Error ↓ | Params (M) | FLOPs | GPU Days ↓ |
|---|---|---|---|---|
| Tiny (6M) | | | | |
| AutoFormer (Chen et al., 2021b) | 25.3 | 5.7 | 1.30G | 24 |
| AZ-NAS (Lee & Ham, 2024) | 23.9 | 5.9 | 1.38G | 0.03 |
| **Dextr (ours)** | **23.7** | 5.8 | 1.36G | 0.03 |
| Small (24M) | | | | |
| AutoFormer (Chen et al., 2021b) | 18.3 | 22.9 | 5.10G | 24 |
| TF-TAS (Zhou et al., 2022) | 18.1 | 23.9 | 5.16G | 0.5 |
| AZ-NAS (Lee & Ham, 2024) | **17.8** | 23.8 | 5.13G | 0.07 |
| **Dextr (ours)** | 18.0 | 23.0 | 4.93G | 0.07 |

et al., 2020b), PC-DARTS (Xu et al., 2019), P-DARTS (Chen et al., 2019), L$^2$-NAS (Mills et al., 2021), $\beta$-DARTS (Ye et al., 2022), PRE-NAS (Peng et al., 2022), TE-NAS (Chen et al., 2021c), MeCo (Jiang et al., 2023), and ZiCo (Li et al., 2023).

Next, we consider our evaluation in the AutoFormer search space (Chen et al., 2021b). The AutoFormer (Chen et al., 2021b) search space is segregated into three subspaces- tiny, small and base with varying architecture sizes. In our experiments, we consider the tiny and small subspaces of AutoFormer to leverage faster training times. We follow the standard protocol (Chen et al., 2021b; Zhou et al., 2022; Lee & Ham, 2024) which involves searching for the parameter-constrained architecture with the best Dextr score among 1000 candidate architectures. The selected ViT is then trained on ImageNet with an identical experimental setup as (Zhou et al., 2022; Lee & Ham, 2024). Our approach is compared with several baselines, namely AutoFormer (Chen et al., 2021b), TF-TAS (Zhou et al., 2022), and AZ-NAS (Lee & Ham, 2024). We compare the top-1 error, number of parameters (Params), number of floating point operations (FLOPs) and search cost of the architecture in GPU days found through our method with the baselines.

**Discussion of Results** Our NAS experiments on DARTS and AutoFormer search spaces are summarised in Table 2 and 3 respectively. Table 2 shows that our proposed approach efficiently searches for architectures with low test errors on ImageNet, completing the search in only 0.07 GPU days. Specifically, Dextr achieves lower error rates and is faster compared to previous one-shot methods, such as DARTS (Liu et al., 2019), SNAS (Xie et al., 2018), and PC-DARTS (Xu et al., 2019), while relying on only a single label-free data sample. Furthermore, Dextr demonstrates superior performance over all prior state-of-the-art zero-shot methods with a comparable search cost. Although some one-shot approaches like FairDARTS (Chu et al., 2020b), P-DARTS (Chen et al., 2019), $\beta$-DARTS (Ye et al., 2022), and PRE-NAS (Peng et al., 2022) achieve better test error, our approach is at least approximately 2.3x faster than any of these approaches with marginal difference in test error. This speedup makes our proxy particularly useful in practical scenarios with limited compute.

Moreover, we observe from Table 3 that our approach outperforms the previous best methods in the AutoFormer-tiny search space and demonstrates competitive performance in the AutoFormer- small search space in terms of Top-1 error while being at least as fast as the previous fastest approach.

### 4.3 Ablation Studies

### 4.3.1 Stability of Dextr

---

[†]Experiments marked by [†] are rerun through their official implementation using the NAS-Bench-Suite Zero (Krishnakumar et al., 2022) framework.

Table 4: Comparison of Spearman Rank Correlation Coefficient ($\rho$) using individual characteristics with Zen and ZiCo on NAS-Bench-201, along with the comparison of Dextr with the harmonic mean of Zen and ZiCo. Here, C/G refers to convergence/generalisation and E refers to expressivity.

| Component | Property Type | CIFAR-10 | CIFAR-100 | ImageNet16-120 |
|---|---|---|---|---|
| $\log(1 + c(\mathbf{X}))$ | C/G | **0.88** | **0.89** | **0.85** |
| ZiCo Li et al. (2023) | C/G | 0.76 | 0.79 | 0.77 |
| $\log(1 + \kappa)$ | E | **0.51** | **0.49** | **0.51** |
| Zen Lin et al. (2021) | E | 0.33 | 0.34 | 0.38 |
| HM(Zen, ZiCo) | C/G + E | 0.44 | 0.44 | 0.47 |
| **Dextr** | C/G + E | **0.90** | **0.91** | **0.87** |

In this study, we assess the stability of our approach. To this end, we sample 10 architectures from NAS-Bench-201 (Dong & Yang, 2020) search space, ranging from 50% to 90% accuracy. Next, for each network, we calculate 10 Dextr values corresponding to 10 random inputs taken from CIFAR-10, along with randomly generated circular inputs $\mathbf{g}(\theta)$ for our curvature calculation. Finally, we calculate the mean and standard deviation of the different Dextr values of each network and report them in Figure 2, along with the test accuracy of that network.

As shown in Figure 2, our approach exhibits strong stability across random CIFAR-10 inputs and randomly sampled circular inputs. The standard deviation of Dextr across multiple input samples remains consistently low for all 10 networks (ranging between 0.01 and 0.05), indicating that the proxy is robust to variations in input data and curvature sampling. Furthermore, the mean Dextr scores correlate well with test accuracy, i.e. networks with higher Dextr values achieve higher accuracy, while those with lower Dextr values perform worse. Notably,

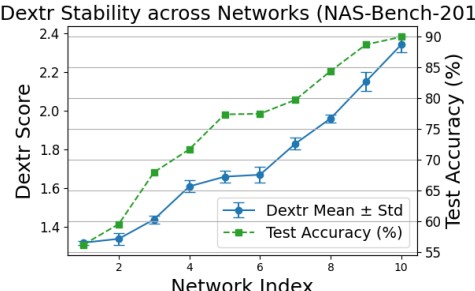

Figure 2: Mean and Standard Deviation of Dextr for 10 networks (sorted by accuracy) sampled from NAS-Bench-201 (Dong & Yang, 2020), corresponding to 10 random inputs taken from CIFAR-10 and 10 randomly generated circular inputs $\mathbf{g}(\theta)$.

the best-performing network (Network with 89.90% accuracy) has the highest mean Dextr score (2.34), whereas the lowest-performing network (Network with 56.19% accuracy) has the lowest Dextr value (1.32). The mid-range networks show that Dextr is able to separate similarly performing architectures, which suggests that Dextr is sensitive enough to capture subtle performance differences between architectures.

### 4.3.2 Performance of individual components of Dextr

We now compare the Spearman rank correlation coefficient $\rho$ of individual components of Dextr, i.e. the condition number term $\log(1 + c(\mathbf{X}))$ (focused on convergence/generalisation) and the curvature term $\log(1 + \kappa)$ (focused on expressivity) with previous proxies considering these individual characteristics on NAS-Bench-201. We consider Zen (Lin et al., 2021) as the expressivity proxy and ZiCo (Li et al., 2023) as the convergence/generalisation proxy. Additionally, we also compare Dextr with a simple harmonic mean of these proxies. The results can be observed in Table 4.

The results demonstrate that Dextr, which combines convergence/generalisation and expressivity components, consistently achieves the highest Spearman rank correlation across all three datasets in NAS-Bench-201. Notably, the condition number term $\log(1 + c(\mathbf{X}))$ outperforms ZiCo across all tasks, indicating that Dextr's convergence/generalisation measure is a more reliable proxy for performance than ZiCo. Moreover, the curvature term $\log(1 + \kappa)$ also consistently outperforms Zen, indicating its effectiveness. Lastly, the harmonic mean of Zen and ZiCo performs worse than either of Dextr's individual components and significantly worse than Dextr itself, suggesting that naively combining external proxies is not as effective as Dextr.

### 4.3.3 Analysis for individual layers

We study the variation in linear independence of feature maps (FMI) across different hidden layers of networks sampled from NAS-Bench-201 (Dong & Yang, 2020). The FMI is calculated using the inverse of the condition number, i.e. $1/c(\mathbf{X}^\phi)$. We compare the FMI between layers of a high-performing network, with a validation accuracy of 89.43% and a low-performing network, with a validation accuracy of 63%. The results, shown in Figure 3, reveal that the FMI of the networks fluctuates due to considering outputs from all layers, including activations, convolutions, and fully connected layers. Peaks in FMI increase, then slightly decrease, and finally drop to zero, with the highest peaks occurring during network dimensionality expansion. This is because dimensionality expansion increases the rank capacity of the feature matrix, allowing it to span

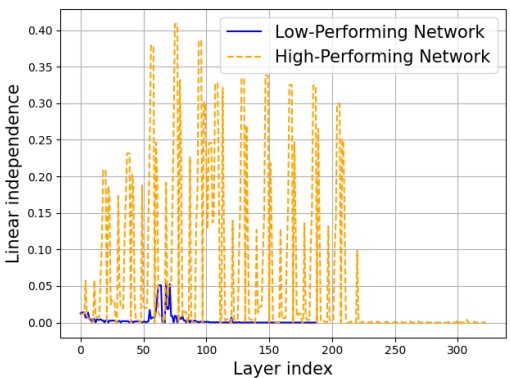

Figure 3: Graph depicting the linear independence of outputs, calculated using $1/c(\mathbf{X}^\phi)$, for each hidden layer of two CNNs. We observe that the hidden layer outputs of the high-performing network are more linearly independent than the low-performing network.

a larger subspace with more orthogonal directions. As a result, the collinearity between feature channels is reduced, and feature linear independence (FMI) peaks. Later, when the network compresses feature dimensionality, the rank of the feature matrix decreases, which squeezes the singular value spectrum and FMI drops. Finally, we observe that the low-performing network's layer outputs are generally less linearly independent, supporting the theoretical findings in Theorems 3 and 4.

## 5 Conclusion

We presented a zero-cost proxy, Dextr. To this end, we first demonstrated the relationship between the collinearity of the features and the convergence and generalisation of a network through SVD. Next, we exploited concepts from Riemannian geometry to include expressivity in our proxy. Our experiments demonstrated that our proxy exhibits a strong correlation with network performance without requiring labelled data. As a result, it outperforms all previous proxies in NAS-Bench-101, NAS-Bench-201, and TransNAS-Bench-micro benchmarks, along with established multi-shot and zero-shot NAS baselines in the ImageNet experiment in the DARTS and AutoFormer search space, all while inducing a low search cost.

## Acknowledgement

We are thankful to the German National Science Foundation (DFG), which supported and funded this work under the project *Always-on Deep Neural Networks* (grant number BE 7212/7-1 — OR245/19-1). Additionally, we acknowledge the computational resources provided by the Erlangen National High Performance Computing Center (NHR@FAU) of the Friedrich-Alexander-Universität Erlangen-Nürnberg (FAU).

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

# A  Appendix

## A.1  Proofs and Derviations

### A.1.1  Proof for Theorem 3

We start from Theorem 1, which states:

**Theorem 1** *Let the input be $i \in [n]$, $c_{low} < \|\mathbf{x}_i\|_2 < c_{high}$ and $\|y_i\| < C$, where $C$ is a constant. If the number of nodes $m$ is set to $\Omega(\frac{n^6}{\lambda_{min}(\mathbf{H}^\infty)^4 \delta^6})$, where $\delta$ is the probability of failure, $\Omega$ denotes the lower bound, and $\lambda_{min}(\mathbf{H}^\infty)$ denotes the scaled minimum eigenvalue of $\mathbf{H}^\infty$, and we i.i.d. initialise $\mathbf{w}_r \sim \mathcal{N}(\mathbf{0}, \mathbf{I})$ and $a_r \sim \mathcal{U}\{-1, 1\}$, where $\mathcal{U}\{\cdot\}$ represents the uniform distribution, then with at least $1 - \delta$ probability, the following relationship holds,*

$$\|\mathbf{u}_i(t) - \mathbf{y}_i\|_2^2 \leq \exp(-\lambda_{min}(\mathbf{H}^\infty)t)\|\mathbf{u}_i(0) - \mathbf{y}_i\|_2^2. \tag{9}$$

By relying on Lemma 1, we know that $\sigma_{max}(\mathbf{X}^\phi(t = 0)) \geq 1$ and thus, $\lambda_{\max}(\mathbf{H}^\infty) \geq 1$. Hence, we divide the term $\lambda_{min}(\mathbf{H}^\infty)t$ with $\lambda_{\max}(\mathbf{H}^\infty)$ and obtain the following inequality

$$\|\mathbf{u}_i(t) - \mathbf{y}_i\|_2^2 \leq \exp\left(-\frac{\lambda_{\min}(\mathbf{H}^\infty)}{\lambda_{\max}(\mathbf{H}^\infty)}t\right)\|\mathbf{u}_i(0) - \mathbf{y}_i\|_2^2. \tag{10}$$

This inequality holds because $\lambda_{\max}(\mathbf{H}^\infty) \geq 1$ implies that $\frac{\lambda_{\min}(\mathbf{H}^\infty)}{\lambda_{\max}(\mathbf{H}^\infty)} \leq \lambda_{\min}(\mathbf{H}^\infty)$, and the exponential function with negative exponent is monotonically decreasing.

Next, we use Remark 3 to approximate the gram matrix concerning the fully connected layer of $f$, i.e. $\mathbf{H}^\infty$ with the gram matrix of the convolutional layer of the CNN $f_{cnn}$ $\mathbf{H}^\phi(t = 0)$. This allows us to re-parameterize the eigenvalues $\lambda_k$ of $\mathbf{H}^\infty$ into singular values $\sigma_k(\mathbf{X}^\phi(t = 0))$ using the equation $\sigma_k(\mathbf{X}^\phi(t)) = \sqrt{\lambda_k(\mathbf{H}^\phi(t))}$. Then, by replacing the network output of $f$, i.e. $\mathbf{u}_i(t)$ with the network output of $f_{cnn}$, i.e. $\mathbf{u}_i^\phi(t)$, we obtain

$$\|\mathbf{u}_i^\phi(t) - \mathbf{y}_i\|_2^2 \leq \exp\left(-\left(\frac{\sigma_{\min}(\mathbf{X}^\phi(0))}{\sigma_{\max}(\mathbf{X}^\phi(0))}\right)^2 t\right)\|\mathbf{u}_i^\phi(0) - \mathbf{y}_i\|_2^2$$

$$\leq \exp\left(-\frac{t}{c\left(\sigma_{\max}(\mathbf{X}^\phi(0))\right)^2}\right)\|\mathbf{u}_i^\phi(0) - \mathbf{y}_i\|_2^2. \tag{11}$$

which completes the proof.

### A.1.2  Proof for Theorem 4

We start from from Theorem 2, which states

**Theorem 2** *Let the loss function of $f$ with $m \to \infty$ evaluated on the test set be denoted as $L(\mathbf{W})$. Let the ground truth $\mathbf{y} = (y_1, ..., y_N)^T$, and $\gamma$ represent the step size of stochastic gradient descent (SGD), determined by $\gamma = kC_1\sqrt{\mathbf{y}^T(\mathbf{H}^\infty)^{-1}\mathbf{y}}/(m\sqrt{N})$, where $k$ is a sufficiently small absolute constant. If Theorem 1 holds, then for any $\delta \in (0, e^{-1}]$, there exists a value $m^*$ which depends on $\delta, N, \lambda_{min}(\mathbf{H}^\infty$ such that if $m \geq m^*$, with a probability of at least $1 - \delta$, the following inequality holds:*

$$\mathbb{E}[L(\mathbf{W})] \leq \mathcal{O}\left(C_2\sqrt{\frac{\mathbf{y}^T\mathbf{y}}{\lambda_{min}(\mathbf{H}^\infty)N}}\right) + \mathcal{O}\left(\sqrt{\frac{\log(1/\delta)}{N}}\right) \tag{12}$$

, where $C_1$ and $C_2$ are constants.

Using Lemma 1, we know that $|\sigma_{max}(\mathbf{X}^\phi(t = 0))| \geq 1$ and thus, $\lambda_{\max}(\mathbf{H}^\infty) \geq 1$. Thus, we can derive the following inequality:

$$\mathbb{E}[L(\mathbf{W})] \leq \mathcal{O}\left(C_2\sqrt{\frac{\lambda_{\max}(\mathbf{H}^\infty)}{\lambda_{\min}(\mathbf{H}^\infty)}}\sqrt{\frac{\mathbf{y}^T\mathbf{y}}{N}}\right) \tag{13}$$

$$+\mathcal{O}\left(\sqrt{\frac{\log(1/\delta)}{N}}\right) \tag{14}$$

Next, we use Remark 3 to approximate the gram matrix of a fully connected layer $\mathbf{H}^\infty$ with a convolution layer $\mathbf{H}^\phi(t=0)$. This allows us to re-parameterize the eigenvalues $\lambda_k(\mathbf{H}^\infty)$ into singular values $\sigma_k(\mathbf{X}^\phi(t=0))$ using the equation $\sigma_k(\mathbf{X}^\phi(t=0)) = \sqrt{\lambda_k \mathbf{H}^\phi(t=0)}$. By replacing the loss term $\mathbb{E}[L(\mathbf{W})]$ with the CNN loss term $\mathbb{E}[L^\phi(\mathbf{W})]$, we obtain

$$\mathbb{E}[L^\phi(\mathbf{W})] \leq \mathcal{O}\left(C_2\frac{\sigma_{\max}(\mathbf{X}^\phi(0))}{\sigma_{\min}(\mathbf{X}^\phi(0))}\sqrt{\frac{\mathbf{y}^T\mathbf{y}}{N}}\right)$$

$$+ \mathcal{O}\left(\sqrt{\frac{\log(1/\delta)}{N}}\right)$$

$$\leq \mathcal{O}\left(C_2 c(\mathbf{X}^\phi(0))\sqrt{\frac{\mathbf{y}^T\mathbf{y}}{N}}\right)$$

$$+ \mathcal{O}\left(\sqrt{\frac{\log(1/\delta)}{N}}\right). \tag{15}$$

which completes the proof.

### A.1.3 Proof for Lemma 1

Let $x_{\hat{i}\hat{j}}$ denote the $\hat{i}\hat{j}$-th entry of $\mathbf{X}^\phi(t)$. We know from the Courant–Fischer–Weyl Min-Max-theorem (Il'yasov & Muravnik, 2022) that $\sigma_{max}(\mathbf{X}^\phi(t)) = \max_{\|u\|=1}\|\mathbf{X}^\phi(t)u\|)$ or any unit vector $u$. Then, we obtain

$$\sigma_{max}(\mathbf{X}^\phi(t) \geq \sqrt{\sum_{i=1}^C x_{\hat{i}\hat{j}}^2} \tag{16}$$

$$\geq \sqrt{x_{\hat{i}\hat{j}}^2} \tag{17}$$

$$\geq |x_{\hat{i}\hat{j}}| \forall \hat{i}, \hat{j} \tag{18}$$

, which shows that the maximum singular value of $\mathbf{X}^\phi(t)$ is larger than the maximum value in the matrix $\mathbf{X}^\phi(t)$. Next, we assume that at least one absolute entry in the output induced by the ReLU activation function is greater than 1, i.e. $|x_{\hat{i}\hat{j}}| \geq 1 \exists \hat{i}, \hat{j}$. Thus, the lemma $\sigma_{max}(\mathbf{X}^\phi(t)) \geq 1$ holds true in general, which is further validated in Section A.1.4.

### A.1.4 Experimental validation for Lemma 1

To provide experimental validation for Lemma 1, we conduct an analysis examining the number of cases where our Lemma holds true, i.e. where $\sigma_{max}(\mathbf{X}^\phi(t)) \geq 1$. To this end, we sample 1000 networks from NAS-Bench-101 (Ying et al., 2019), NAS-Bench-201 (Dong & Yang, 2020), and NAS-Bench-301 (Zela et al., 2022) through the NAS-Bench-Suite-Zero (Krishnakumar et al., 2022) framework. Next, we input a random data sample from CIFAR-10 to each of these networks and calculate the mean of the percentage of layers

Table 5: The mean percentage of layers corresponding to 1000 networks in three different benchmarks where Lemma 1 holds true, i.e. $\sigma_{max}(\mathbf{X}^{\phi}(t)) \geq 1$.

| Benchmark | $\sigma_{max}(\mathbf{X}^{\phi}(t)) \geq 1$ |
|---|---|
| NAS-Bench-101 | 99.18 % |
| NAS-Bench-201 | 90.01 % |
| NAS-Bench-301 | 99.51 % |

where $\sigma_{max}(\mathbf{X}^{\phi}(t)) \geq 1$. From Table 5, we observe that for all the benchmarks for at least 90% of the cases, $\sigma_{max}(\mathbf{X}^{\phi}(t)) \geq 1$. Remarkably, in NAS-Bench-101 and NAS-Bench-301, Lemma 1 holds true for almost all the layer outputs.

## A.2 Transferability to deep neural networks

We now elaborate on how one can extend the convergence theorems (Theorems 1 and 3) to deep neural networks, relying on the proof by Du et al. (2019b). Consider a deep neural network of the form:

$$f(x, \mathbf{W}, \mathbf{a}) = \mathbf{a}^T \sigma(\mathbf{W}^{(H)} \ldots \sigma(\mathbf{W}^{(1)} \mathbf{x})), \tag{19}$$

where $\mathbf{x} \in \mathbb{R}^d$ is the input and $\mathbf{a} \in \mathbb{R}^m$ is the output layer. If $h = 2, \ldots, H$ and $\mathbf{W}^{(h)} \in \mathbb{R}^{m \times m}$ are the middle layers, then the associated gram matrix $\mathbf{G}^{(h)}(t) \in \mathbb{R}^{n \times n}$ will have values $\mathbf{G}_{ij}^{(h)}(t) = \left\langle \frac{\partial u_i(t)}{\partial \mathbf{W}^{(h)}(t)}, \frac{\partial u_j(t)}{\partial \mathbf{W}^{(h)}(t)} \right\rangle$, where $u_i$ and $u_j$ are the i-th and j-th predictions respectively.

Du et al. (2019b) show that $\sum_{h=1}^{H} \mathbf{G}^{(h)}(t)$ has a lower bounded least eigenvalue because i and j are not parallel. Moreover, $\sum_{h=1}^{H} \mathbf{G}^{(h)}(0)$ is close to the fixed matrix $\sum_{h=1}^{H} \mathbf{G}_{\infty}^{(h)}$ and $\sum_{h=1}^{H} \mathbf{G}^{(h)}(t)$ is close to $\sum_{h=1}^{H} \mathbf{G}^{(h)}(0)$. Therefore, the proof for the convergence theorem by Du et al. (2019b) is transferable to deep neural networks with $H$ hidden layers. Moreover in this case, the rate of convergence in a multi-layer case is governed by $\lambda_{min}(\sum_{h=1}^{H} \mathbf{G}_{\infty}^{(h)})$.

Now, using Weyl's inequality for Hermitian matrices, we know that

$$\sum_{h=1}^{H} \lambda_{min} \mathbf{G}_{\infty}^{(h)} \leq \lambda_{min}(\sum_{h=1}^{H} \mathbf{G}_{\infty}^{(h)}). \tag{20}$$

Through this inequality, we can observe that the sum of individual minimum eigenvalues of each layer's Gram matrix provides a lower bound on the total minimum eigenvalue governing convergence. Adding negative and exponential on both sides, we know that

$$-\exp(\sum_{h=1}^{H} \lambda_{min} \mathbf{G}_{\infty}^{(h)}) \geq -\exp(\lambda_{min}(\sum_{h=1}^{H} \mathbf{G}_{\infty}^{(h)})). \tag{21}$$

Hence, $\sum_{h=1}^{H} \lambda_{min} \mathbf{G}_{\infty}^{(h)}$ can be replaced with $\lambda_{min}(\sum_{h=1}^{H} \mathbf{G}_{\infty}^{(h)})$ in the convergence theorem for multi-layer networks. Hence, $\sum_{h=1}^{H} \lambda_{min} \mathbf{G}_{\infty}^{(h)}$, utilised by Dextr, also governs the convergence rate of a deep neural network. Therefore, Theorems 1 and 3 are transferable to deep neural networks with $H$ hidden layers.

## A.3 Detailed approximation to convolutional layer

We now provide a detailed approximation of a single-layer multi-sampled fully connected layer to a convolutional layer. Jiang et al. (2023) show that the convolutional layer operation with an input $x_{in} \in \mathbb{R}^{c_{in} \times w \times h}$ and a filter $w \in \mathbb{R}^{c_{in} \times k \times k}$, with a stride of one and zero padding, can formally be expressed as

$$[x_{out}]_{i,j} = \sum_{c=1}^{c_{in}} \sum_{a=-p}^{p} \sum_{b=-p}^{p} [w^c]_{a+p+1,b+p+1} \cdot [x_{in}^c]_{a+i,b+j},$$

Table 6: Comparison of the Spearman rank correlation coefficient on NATS-Bench-SSS (Dong et al., 2021) across different datasets for various zero-cost proxies.

| Methods | CIFAR-10 | CIFAR-100 | ImageNet16-120 |
|---|---|---|---|
| #params | 0.72 | 0.73 | 0.84 |
| Fisher (Turner et al., 2020) | 0.44 | 0.55 | 0.47 |
| Snip (Lee et al., 2019) | 0.59 | 0.62 | 0.76 |
| Grasp (Wang et al., 2020) | -0.13 | 0.01 | 0.42 |
| Synflow (Tanaka et al., 2020) | 0.81 | 0.80 | 0.57 |
| Grad_Norm (Abdelfattah et al., 2021) | 0.51 | 0.49 | 0.67 |
| NTK (Chen et al., 2021c) | 0.34 | 0.29 | 0.28 |
| Zen (Lin et al., 2021) | 0.69 | 0.71 | 0.87 |
| ZiCo (Li et al., 2023) | 0.73 | 0.75 | 0.88 |
| MeCo (Jiang et al., 2023) | -0.79 | -0.87 | -0.86 |
| **Dextr (ours)** | -0.63 ± 0.006 | -0.60 ± 0.005 | -0.67 ± 0.006 |

where $x_{in}^c$ and $w^c$ represent the c-th channel of $x_{in}$ and $w$ respectively, $p = (k-1)/2$, and $i \in [w]$, $j \in [h]$. Next, $x_{out}$ and each input channel $x_{in}^c$ can be flattened into one-dimensional vectors, $\tilde{x}_{out} \in \mathbb{R}^{d \times 1}$ and $\tilde{x}_{in}^c \in \mathbb{R}^{d \times 1}$ respectively, where $d = w \times h$. This transformation leads to the following approximation, viewing the convolutional layer as a multi-sample fully-connected network:

$$\tilde{x}_{out} = \sum_{c=1}^{c_{in}} A_c \sigma((B_c \circ W)^T \tilde{x}_{in}^c),$$

subject to the following constraints:

$$A_c \in \mathbb{R}^{d \times d_h}, [A_c]_{ij} = \mathbb{I}\{j = (c-1)d + i\},$$

$$B_c \in \mathbb{R}^{d_h \times d}, [B_c]_{ij} = \mathbb{I}\{(c-1)d < i \le cd\},$$

$$B_c \circ W \text{ satisfying weight sharing constraints.}$$

Here, $d_h = c_{in} \times d$ and $W \in \mathbb{R}^{d \times d_h}$ is the weight matrix. We can further simplify this by relaxing the second constraint to $B_c = 1_{d_h \times d}$ and ignore the last constraint. As a result, we view each flattened input channel $\tilde{x}_{in}^c$ as an independent data sample, forming a multi-sample fully-connected network.

### A.4 Additional Experiments

### A.5 NATS-Bench-SSS

We now conduct correlation experiments on the NATS-Bench-SSS (Dong et al., 2021) benchmark. The NATS-Bench-SSS benchmark is fundamentally different from other correlation benchmarks (Dong & Yang, 2020; Ying et al., 2019; Zela et al., 2022), as it contains architectures of varying numbers of channels. We follow the same evaluation protocol as the other correlation experiments, detailed in Section 4.1 of the main paper. We report the mean Spearman Rank Correlation Coefficient $\rho$ and standard deviation, averaged over 3 runs of Dextr in Table 6 and compare it with various baselines.

We observe from the results that in unlike other benchmarks, our approach exhibits a negative correlation weaker than some of the previous methods in the NATS-Bench-SSS benchmark. Upon investigating, we found out that this discrepancy arises due to the sensitivity of the condition number $c(\mathbf{X})$ to the number of channels, similarly exhibited by MeCo (Jiang et al., 2023). This behaviour is specific to the NATS-Bench-SSS benchmark as it contains architectures of varying number of channels. We rectify this behaviour for this benchmark through the optimisation procedure detailed in the Section A.5.1.

Table 7: Comparison of the Spearman rank correlation coefficient between Dextr and the optimised version of Dextr (Dextr$_{opt}$) on NATS-Bench-SSS (Dong et al., 2021).

| Methods | CIFAR-10 | CIFAR-100 | ImageNet16-120 |
|---------|----------|-----------|----------------|
| Dextr | **-0.63** | -0.60 | **-0.67** |
| Dextr$_{opt}$ | 0.56 | **0.66** | 0.65 |

### A.5.1   Dextr optimisation

As observed in the NATS-Bench-SSS (Dong et al., 2021) correlation experiment detailed in Table 6, Dextr achieves a negative correlation with the accuracy of the architecture. We utilise the optimisation procedure detailed by (Jiang et al., 2023) to fix this discrepancy. Specifically, instead of considering all the channels of the network layer, we randomly sample a fixed number of channels and calculate the Dextr value using the formula

$$
\text{Dextr}_{opt} = \frac{\log\left(1 + \sum_{l=1}^{L} \frac{\alpha_l}{\beta c_l(\mathbf{X}^{\phi'})}\right) \cdot \log(1 + \kappa(\theta))}{\log\left(1 + \sum_{l=1}^{L} \frac{\alpha_l}{\beta c_l(\mathbf{X}^{\phi'})}\right) + \log(1 + \kappa(\theta))}
\tag{22}
$$

, where $\alpha_l$ is the total number of channels of the layer, $\beta$ is the number of sampled channels, and $\mathbf{X}^{\phi'}$ is the concatenated feature map matrix using the sampled channels. This formulation restricts the dimension of the feature map matrix $\mathbf{X}^{\phi'}$ and eliminates the effect caused by the varying number of channels. We present the results of this optimisation procedure on the NATS-Bench-SSS benchmark in the Table 7. As observed, while Dextr$_{opt}$ fixes the discrepancy for the NATS-Bench-SSS (Dong et al., 2021) benchmark, it fails to outperform the original formulation of Dextr in two out of three datasets.

### A.5.2   MobileNet-v2

Our experiment on the MobileNet-v2 (Sandler et al., 2018) search space follows the standard evaluation protocol (Li et al., 2023; Lin et al., 2021; Lee & Ham, 2024) of searching for an architecture through evolutionary algorithm with FLOPs constraint of 450M. We use the same experimental settings as (Lee & Ham, 2024) with a population size of 1024 and a number of iterations as 1e5. Next, we train the selected architecture on ImageNet and report the floating point operations (FLOPs), Top-1 accuracy, and search cost in GPU Days in Table 8. We observe from the results that our method marginally improves over the previous best method AZ-NAS with a smaller searched model.

### A.6   Implementation Details

We now elaborate upon the procedure for the calculation of the Dextr value. First, we input a label-free data sample, i.e. an image from the CIFAR-10 dataset to a network from the respective benchmark and perform one forward pass with the image. In the forward pass, we calculate the sum of the inverse of the condition number of the layer output for each layer, i.e. $(\sum_{l=1}^{L} 1/c_l(\mathbf{X}^{\phi}))$. Next, we obtain the extrinsic curvature $\kappa(\theta)$ by generating a random circular input and then calculating the velocity and acceleration vectors (Eq. 5). Finally, we calculate the Dextr value through Eq. 8.

### A.7   Experimental settings

We run all the search procedures on a single NVIDIA RTX A6000 GPU with 48GB memory. For our correlation experiments on NAS-Bench-101 (Ying et al., 2019), NAS-Bench-301 (Zela et al., 2022), and TransNASBench-101 (Duan et al., 2021), we utilise the same experimental configuration as NAS-Bench-Suite-Zero (Krishnakumar et al., 2022), SWAPNAS (Peng et al., 2024). For our correlation experiments on

Table 8: Quantitative comparison for the MobileNet-v2 (Sandler et al., 2018) search space with FLOPs constraint of 450M. We report the top-1 accuracy, floating point operations (FLOPs) of selected networks on ImageNet (Deng et al., 2009), the type of search method- multi-shot (MS), one-shot (OS) or zero-shot (ZS), along with the search cost in GPU days. The best approach is indicated as **bold** and the second-best approach is indicated as underlined.

| Method | FLOPs (M) | Top-1 Acc. (%) ↑ | Type | Search Cost (GPU Days)↓ |
|---|---|---|---|---|
| NAS-Net-B (Zoph et al., 2018) | 488 | 72.8 | MS | 1800 |
| CARS-D (Yang et al., 2020b) | 496 | 73.3 | MS | 0.4 |
| BN-NAS (Chen et al., 2021a) | 470 | 75.7 | MS | 0.8 |
| OFA (Cai et al.) | 406 | 77.7 | OS | 50 |
| RLNAS (Zhang et al., 2021) | 473 | 75.6 | OS | - |
| DONNA (Moons et al., 2021) | 501 | 78.0 | OS | 405 |
| #params | 451 | 63.5 | ZS | 0.02 |
| ZiCo (Li et al., 2023) | 448 | 78.1 | ZS | 0.4 |
| AZ-NAS (Lee & Ham, 2024) | 462 | 78.6 | ZS | 0.4 |
| **Dextr (ours)** | 457 | **78.8** | ZS | 0.6 |

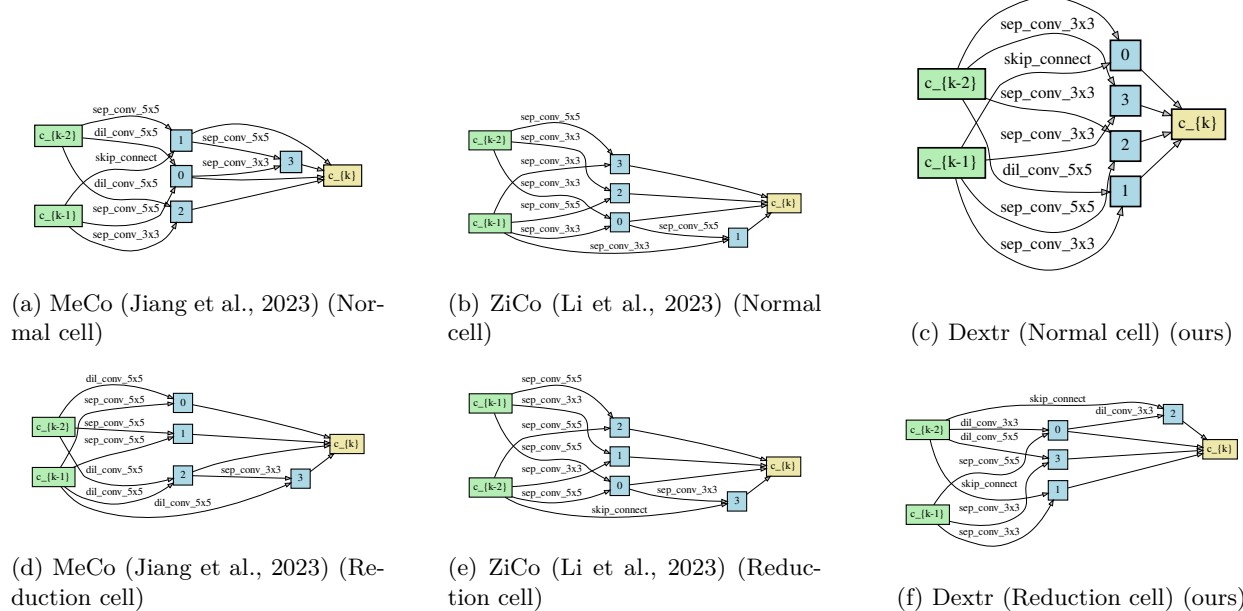

(a) MeCo (Jiang et al., 2023) (Normal cell)

(b) ZiCo (Li et al., 2023) (Normal cell)

(c) Dextr (Normal cell) (ours)

(d) MeCo (Jiang et al., 2023) (Reduction cell)

(e) ZiCo (Li et al., 2023) (Reduction cell)

(f) Dextr (Reduction cell) (ours)

Figure 4: Found architectures from three zero cost proxies in the DARTS (Liu et al., 2019) search space for the ImageNet experiment using Zero-Cost-PT (Xiang et al., 2023).

NAS-Bench-201, we utilize the experimental setup as MeCo (Jiang et al., 2023). Moreover, the experimental configuration of the search on DARTS search space (Liu et al., 2019) using Zero-Cost-PT (Xiang et al., 2023) algorithm is detailed in Table 9. The training on ImageNet for DARTS search space follows the same settings and protocol as (Lukasik et al., 2022; Asthana et al., 2024; Chen et al., 2021c). Lastly, for our AutoFormer and MobileNet-v2 experiments, we utilize the same experimental framework as TF-TAS (Zhou et al., 2022) and AZ-NAS (Lee & Ham, 2024).

Table 9: Experimental settings for search in DARTS search space through Zero-Cost-PT algorithm.

| Settings | Search settings |
|---|---|
| Batch Size | 1 |
| Cutout | False |
| Cutout Length | - |
| Learning Rate | 0.025 |
| Learning Rate Min | 0.001 |
| Momentum | 0.9 |
| Weight Decay | 3e-4 |
| Grad Clip | 5 |
| Init Channels | 16 |
| Layers | 8 |
| Drop Path Prob | - |

## A.8 Visual Comparison

Figure 4 presents a visual comparison of the architectures generated by our proposed proxy, Dextr, alongside those produced by prior works, specifically MeCo (Jiang et al., 2023) and ZiCo (Li et al., 2023). The normal cell (used for feature extraction) generated by Dextr demonstrates a broad range of operations, including skip connections, separable convolutions, and dilated convolutions, while maintaining a densely connected structure. In contrast, the normal cell produced by MeCo exhibits simpler connections, which limits the network's expressivity. Similarly, although the normal cell generated by ZiCo features dense connections, it employs a less diverse set of operations compared to Dextr.

Furthermore, the reduction cell (used for downsampling) generated by Dextr incorporates a more diverse set of operations. It effectively balances feature reuse through skip connections and flexible transformations utilizing dilated and separable convolutions. Thus, this design is particularly well-suited for complex tasks that require robust hierarchical representations. Lastly, we observe that the network selected by Dextr is shallower than other approaches mentioned, which might seem counterintuitive, as it suggests the network does not have enough expressivity. However, since we do not characterise expressivity by explicitly maximising depth (or minimising width), we argue that in this case, the found network from Dextr still maintains adequate expressivity through its choice of operations. To prove this claim, we measure the expressivity by comparing the curvature of the networks presented in Figure 4. We empirically find that the architecture found through our approach has more curvature ($1.2 \times 10^8$) than the architectures found through ZiCo (curvature $= 7.7 \times 10^7$) and MeCo (curvature$= 5.8 \times 10^7$). This implies that even if the chosen network by Dextr does not exhibit the deepest structure, it still is more expressive than the networks found through ZiCo and MeCo, due to its chosen operations, which proves the effectiveness of Dextr in capturing expressivity. This indicates that while deeper networks often exhibit greater expressive capacity, depth alone is not a sufficient indicator., i.e. there can be networks which are shallow, yet expressive, as the one depicted in Figure 4. That is also the reason why, instead of utilising depth directly, we utilise a more reliable metric used in the literature (Poole et al., 2016; Chen et al., 2023), i.e. curvature of the output.

## A.9 Visualisation of correlation

We provide scatter plots between the Dextr score and the validation accuracy of the network across multiple correlation benchmarks, including NAS-Bench-101, NAS-Bench-201, NAS-Bench-301, and TransNAS-Bench-101-micro in Figure 5. All the plots reveal a strong positive correlation between the Dextr score and the performance of the network.

## A.10 Limitations and Future Work

We now discuss some limitations of our work. Firstly, although our proxy does not require labelled data for computation, it still needs to perform slightly expensive calculations of the derivative of the output

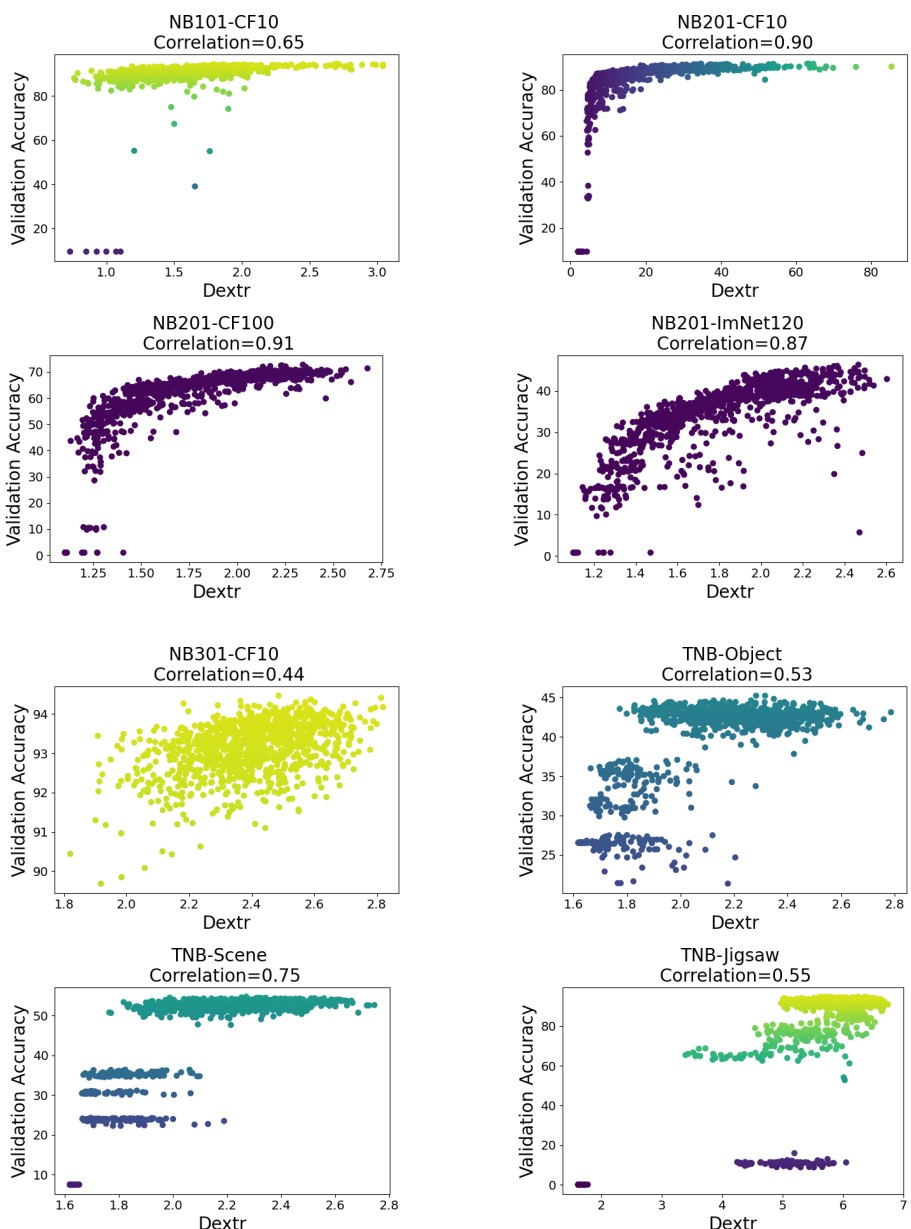

Figure 5: Scatter plots between Dextr value and validation accuracy for NAS-Bench-101 (Ying et al., 2019), NAS-Bench-201 (Dong & Yang, 2020), NAS-Bench-301 (Zela et al., 2022), and TransNAS-Bench-101-micro (Duan et al., 2021)

to compute the velocity $\mathbf{v}(\theta)$ and acceleration $\mathbf{a}(\theta)$ vectors, essential to calculate the expressivity of the network. Future works can focus on this issue by utilising a different characterisation of expressivity in the approach. Lastly, our theoretical analysis only applies to ReLU and GeLU networks. Extending the theoretical framework to other activation functions such as Sigmoid and Tanh can be an important future work direction. Lastly, Lemma 1 requires at least one absolute entry in $\mathbf{X}^\phi$ to be greater than 1. Although this is generally the case, some uncommon data normalisation tecnhniques can falsify this assumption.

