# OpenReview forum: "Dextr: Zero-Shot Neural Architecture Search with Singular Value Decomposition and Extrinsic Curvature"
_TMLR — Accepted by TMLR_

### Review · Reviewer_qJ1j · 2025-06-02

**Summary Of Contributions:**

This paper introduces Dextr, a Zero-Cost Proxy (ZCP) for Neural Architecture Search (NAS) that aims to balance convergence (neural network ability to reach a loss objective), generalisation (performance on a test set) and expressivitity (ability to model complex function). Dextr has a rigorious theoretical grounding in Singular Value Decomposition (SVD) and Riemannian geometry. Further, Dextr is evaluated on multiple benchmarks including NAS-Benchmarks, AutoFormer [1] and MobileNets [2] to demonstrate its veracity. Finally, the authors provide some ablation studies of their approaches.

**Audience:**

Yes

**Broader Impact Concerns:**

N/A
Overall, this is a well-written manuscript with mostly solid results though missing some comparisons and in need of some writing tune-up in places, specifically Section 4.

References:

[1] "Autoformer: Searching transformers for visual recognition" - CVPR'21.

[2] "Mobilenetv2: Inverted Residuals and Linear Bottlenecks" - CVPR'18.

[3] "NAS-Bench-101: Towards reproducible neural architecture search" - ICML'19.

[4] "How powerful are performance predictors in neural architecture search?" - NeurIPS'21.

[5] "L2NAS: Learning to optimize neural architectures via continuous-action reinforcement learning" - CIKM'21.

**Claims And Evidence:**

Yes

**Requested Changes:**

- The reviewer would ask for a greater elaboration on task/data agnostic and data-based proxies in the 2nd paragraph of the introduction, to help future readers to better understand the demarcation betweeen each of them.
- Figure 1(b). Since this is the first figure mentioned chronologically following the text, it should be the first to appear. Also, revise formatting to make the figures bigger and have larger axis/legend font size.
- Table 1 should be moved to the top of the next page (pg. 9) so as to be closer to where it is actually discussed.
- Table 2: Missing comparison with a transferable RL NAS [5] which has similar search cost/results.
- Sections 4.1 and 4.2 should be revised and interwoven. E.g., first have the paragraph explaining the setup leading to Table 1, then the paragraph that introduces Table 1 and discusses results. Then follow with setup and results discussion for Table 2, then 3, etc. This helps a reader understand the context of how you got the results in a Table.
- Table 4 should either be a wraptable environment or at the top of a page.

**Strengths And Weaknesses:**

Strengths:
- Most of the motivation for the paper is well-grounded. In particular, the reviewer appreciates the writing in the third paragraph of the introduction (where No Free Lunch theorem is brought up). Overall, the writing of this manuscript is smooth.
- The method provides a fast and accurate zero-cost proxy.
- The authors provide rigorous mathematical proofs and explanations to back their method.
- The method is evaluated on a plethora of different NAS environments ranging from cell-based benchmarks [3] to Vision Transformers [1] and others.
- Figure 1(a) is very effective.
- Reviewer appreciates additional results in the appendix they are informative.

Weaknesses:
- The motivation for having a ZCP that covers the "expressivitiy" of a network is not very well established. Convergence and generalisation are straightforward, but expressivity not so much. From the reviewer's point of view, the utility of expressivitity is as a means to improve convergence and generalisation; by modeling more complex functions, the network can better achieve its end-to-end objectives.
- Table 1, there is some concern: Also, there is some concern about the results in this table, e.g., for NB301, no method is able to achieve over 0.5 SRCC. With that in mind, what is the motivation for using a ZCP at all on this benchmark when one could just use a performance predictor [4]? Also, the reviewer is puzzled by some of the TNB results, e.g., on Object, only one ZCP achieves negative SRCC, but most ZCP performance (including Dextr) is quite mediocre, while on Scene, several ZCPs obtain negative SRCC but others achieve above 0.6 and Dextr achieves an impressive 0.75. Can the authors provide any commentary on why this happens?
- Table 2 results are more of a mixed bag. Dextr is finding a larger network in terms of #params that performs worse than that found by Oneshot and Predictor methods. While the cost of Dextr is smaller than these methods, Dextr and some Oneshot/Predictor methods all find a network in less than 1 GPU day. Further, given that the cost of training a neural network on ImageNet is several GPU days, the search cost savings do not justify the downstream performance inefficiencies.
- While ablation studies are provided, the reviewer does not feel that they are well-explained. E.g., it is hard to see a clear pattern in Table 4, while Figure 2 is more readily understood as higher independence -> better performance.

---

> ### Author Response · Authors · 2025-07-10
> **Response to the Reviewer qJ1j (1/2)**
>
> We extend our sincere appreciation to the reviewer for their valuable suggestions and insightful comments. Below, we address the concerns of the reviewer.
>
> > The motivation for having a ZCP that covers the "expressivity" of a network is not very well established...
>
> While convergence and generalisation alone are sufficient to somewhat predict the network performance, the NAS methods utilising solely convergence and generalisation have an inherent bias towards wide network topologies (e.g. by including skip connections) [1]. This is because wider architectures typically exhibit improved signal propagation and gradient flow, which enhance both convergence speed and generalisation. On the contrary, expressive networks favor not wide but deeper network topologies. Hence, the No Free Lunch Theorem states that the best networks are the ones that have a balanced width and depth [1]. To encourage this balance, our method incorporates expressivity as an additional signal in the proxy design. This can also be observed in Figure 4 of the Appendix, where we see that Dextr finds networks with more complex connections, compared to MeCo (a proxy focused on convergence and generalisation), and hence with more expressivity. These qualitative differences also support our hypothesis that including expressivity leads to more balanced and potentially more capable architectures. **We have now improved over the motivation to use expressivity in the introduction (Section 1) of our updated manuscript.**
>
> > Table 1, there is some concern: Also, there is some concern about the results in this table, e.g., for NB301, no method is able to achieve over 0.5 SRCC. With that in mind, what is the motivation for using a ZCP at all on this benchmark when one could just use a performance predictor [4]?
>
> NB301 is known to be a particularly challenging benchmark due to high architectural diversity. The low correlation across all methods (including ours and existing proxies) reflects this difficulty. However, zero cost proxies offer key advantages over performance predictors: they require no training data, are task-agnostic, and generalise without retraining. **We have now included an extended explanation on NB301 results in Section 4.1 of our updated manuscript.**
>
> > Also, the reviewer is puzzled by some of the TNB results, e.g., on Object, only one ZCP achieves negative SRCC, but most ZCP performance (including Dextr) is quite mediocre, while on Scene, several ZCPs obtain negative SRCC but others achieve above 0.6 and Dextr achieves an impressive 0.75. Can the authors provide any commentary on why this happens?
>
> We appreciate the reviewer’s observation. The Object Classification task poses a greater challenge for some zero-cost proxies (including Dextr) because the task relies heavily on semantic visual categories (e.g. identifying sofa, home theater, etc.) making it harder for structural or data-agnostic proxies to correlate with final accuracy. In contrast, the Scene Classification task is an easier task that benefits more from global image cues and spatial hierarchies, which are more aligned with the characteristics that zero-cost proxies like Dextr can capture. This is one reason why Dextr achieves stronger correlation on Scene but performs modestly on Object. In particular, Dextr outperforms other proxies in Scene Classification because it captures the expressive capacity of a network to model global spatial patterns and hierarchical features, both of which are critical for scene understanding. Unlike object classification, which often relies on localized, class-specific cues, scene classification depends more on how well a network integrates broader contextual information across the image. As a result, it is better aligned with the demands of the Scene Classification task compared to proxies focused solely on convergence or generalisation (e.g., MeCo, NASWOT), which may prioritize shallower or wider architectures that perform well on simpler tasks but lack the depth or structure needed for scene-level semantics. **We have now included this explanation in the Section 4.1 of our updated manuscript.**

---

> ### Author Response · Authors · 2025-07-10
> **Response to the Reviewer qJ1j (2/2)**
>
> >Table 2 results are more of a mixed bag. Dextr is finding a larger network in terms of #params that performs worse than that found by Oneshot and Predictor methods. While the cost of Dextr is smaller than these methods, Dextr and some Oneshot/Predictor methods all find a network in less than 1 GPU day. Further, given that the cost of training a neural network on ImageNet is several GPU days, the search cost savings do not justify the downstream performance inefficiencies.
>
> The authors agree that when the evaluation cost (e.g., ImageNet training) dominates, search efficiency may matter less, but in many practical scenarios with limited compute, a faster ZCP search is highly valuable. **We have incorporated this argument in Section 4.2 of our updated manuscript.**
>
> > While ablation studies are provided, the reviewer does not feel that they are well-explained. E.g., it is hard to see a clear pattern in Table 4, while Figure 2 is more readily understood as higher independence -> better performance.
>
> We thank the reviewer for the critique. **We have now converted our Table 4 regarding Stability Analysis into a plot, namely Figure 2, for enhanced readability and easy interpretation of results.** The purpose of Figure 2 is to showcase the stability of Dextr, which is visible by low StD values among all networks. **Moreover, we have elaborated more on all the ablation studies in Section 4.3 of our updated manuscript.**
>
> > The reviewer would ask for a greater elaboration on task/data agnostic and data-based proxies in the 2nd paragraph of the introduction, to help future readers to better understand the demarcation betweeen each of them.
>
> Data-agnostic proxies evaluate an architecture’s potential without using any input data. They rely solely on the network's structure to estimate the network performance. In contrast, data-based proxies estimate network performance by passing a small batch of data through the network. **We have elaborated more on the data-agnostic and data-based proxies in the introduction (Section 1) in our updated manuscript.**
>
>
> **Figure and Table adjustments**
>
> We have made the following Table and Figure adjustments in our updated manuscript:
> - Figure 1(b) now appears first and Figure 1 has revised formatting
> - Table 1 moved to the next page
> - Sections 4.1 and 4.2 are now interwoven based on the reviewers comments.
> - Table 4 is now reported as a plot in Figure 2.
> - We have included the results from the transferable RL NAS method [2] in Table 2 of our manuscript.
>
> **References**
>
> [1] Chen, Wuyang, et al. ‘No Free Lunch in Neural Architectures? A Joint Analysis of Expressivity, Convergence, and Generalization’. AutoML Conference 2023, 2023.
>
> [2] Mills, Keith G., et al. "L2nas: Learning to optimize neural architectures via continuous-action reinforcement learning." Proceedings of the 30th ACM International Conference on Information & Knowledge Management. 2021.

---

> > ### Comment · Reviewer_qJ1j · 2025-07-21
> >
> > The reviewer thanks the authors for their detailed reply and edits to the manuscript, especially with respect to formatting. It is really appreciated. Overall, the reviewer is mostly satisfied with the empirical results of the paper, but less so with the reasoning/motivation behind them.
> >
> > Specifically, while the reviewer thanks the authors for clarifying concerns about motivation, ZCPs, and expressivity, the reviewer does not believe the claims made in the paper are sufficiently aligned with the results produced and the writing should be strengthened to more align with what Dextr produces.
> > Take for instance expressivity, the author's claim: "the NAS methods utilising solely convergence and generalization have an inherent bias towards wide network topologies (e.g. by including skip connections)" which does make sense, this has been documented in the literature [1, 2]. However, the authors then direct towards Fig. 4 in the appendix, where the normal cell found by their method has the widest/shallowest topology compared amongst all shown with every intermediate (0-3) node connecting to both inputs and never to a previous intermediate node. This shows that either Dextr does not achieve adequate expressivity or is a finding that can be used to argue against the value of expressivity.
> >
> > Next, while the reviewer appreciates the additional detail about NB-301 and Object Detection for Trans-NB-101, the reviewer asks if there is any other literature that further reinforces these claims?
> >
> > Finally, Figure 2 is easier to understand now. The reviewer thanks the authors greatly for their revision. However, it does not instill confidence in the method. This is because the Dextr only assigns the highest score to the network with the best accuracy, and the worst score to the network with the worst accuracy, while everything in between is jumbled. Specifically, Dextr erronously assigns the wrong scores to networks 2-5, labeling the worst the best and vis-versa. It almost does the same for networks 6-9. Therefore, the reviewer would urge the authors to try and provide a better visual example to show how the scores of their method align with ground-truth performance.
> >
> > In sum: The reviewer still considers the paper borderline. The main results are solid however the theory/motivation backing them up is mismatched and there are further issues with ablation studies. Therefore, the reviewer withholds a final recommendation for now.

---

> > > ### Comment · Reviewer_qJ1j · 2025-07-22
> > > **Forgot to add references**
> > >
> > > [1] https://arxiv.org/pdf/1909.09569
> > >
> > > [2] https://openreview.net/forum?id=EMys3eIDJ2

---

> > > > ### Author Response · Authors · 2025-07-23
> > > > **Response to the Official Comment by Reviewer qJ1j**
> > > >
> > > > We thank the reviewer for their additional feedback on our paper. Below, we address the concerns:
> > > >
> > > > > Take for instance expressivity, the author's claim: "the NAS methods utilising solely convergence and generalization have an inherent bias towards wide network topologies (e.g. by including skip connections)" which does make sense, this has been documented in the literature [1, 2]. However, the authors then direct towards Fig. 4 in the appendix, where the normal cell found by their method has the widest/shallowest topology compared amongst all shown with every intermediate (0-3) node connecting to both inputs and never to a previous intermediate node. This shows that either Dextr does not achieve adequate expressivity or is a finding that can be used to argue against the value of expressivity.
> > > >
> > > > The authors appreciate the critical observation by the reviewer. The reviewer is right about the given example in the Appendix that the network selected by Dextr is shallower than other approaches mentioned, which might seem counterintuitive, as it suggests the network does not have enough expressivity. However, since we do not characterise expressivity by explicitly maximising depth (or minimising width), we argue that in this case, the found network from Dextr still maintains adequate expressivity through its choice of operations. To prove this claim, we measure the expressivity by comparing the curvature of the networks presented in Figure 4 of the Appendix, which can be observed below.
> > > >
> > > > | MeCo | ZiCo | Dextr (ours) |
> > > > | -------- | -------- | -------- |
> > > > |  5.8 $\times$ 10^7  | 7.7 $\times$ 10^7   |   **1.2 $\times$ 10^8**  |
> > > >
> > > > We observe from the Table that even if the chosen network does not exhibit the deepest structure, it still is more expressive than the networks found through ZiCo and MeCo, due to its chosen operations, which proves the effectiveness of Dextr in capturing expressivity. This indicates that while deeper networks often exhibit greater expressive capacity, depth alone is not a sufficient indicator., i.e. there can be networks which are shallow, yet expressive, as the one depicted in Figure 4. That is also the reason why, instead of utilising depth directly, we utilise a more reliable metric used in the literature [1,2], i.e. curvature of the output. **We have included this explanation in the Appendix Section A.8 of our updated manuscript.**
> > > >
> > > >
> > > > > Next, while the reviewer appreciates the additional detail about NB-301 and Object Detection for Trans-NB-101, the reviewer asks if there is any other literature that further reinforces these claims?
> > > >
> > > > **Thank you for the suggestion. We have now included references to further literature in the claims regarding NB301 and TNB101 in the Section 4.1 of our updated manuscript.**
> > > >
> > > > >  The reviewer thanks the authors greatly for their revision. However, it does not instill confidence in the method. This is because the Dextr only assigns the highest score to the network with the best accuracy, and the worst score to the network with the worst accuracy, while everything in between is jumbled. Specifically, Dextr erronously assigns the wrong scores to networks 2-5, labeling the worst the best and vis-versa. It almost does the same for networks 6-9. Therefore, the reviewer would urge the authors to try and provide a better visual example to show how the scores of their method align with ground-truth performance.
> > > >
> > > > We thank the reviewer for this helpful suggestion. First, we would like to clarify why the scores of the intermediate networks in Figure 2 did not perfectly align with the accuracy, as the reviewer correctly observed. This effect arises due to a low range of accuracies in the sampled networks. In such a narrow band, local misrankings are expected because zero-shot proxies are not designed to be perfect predictors, but rather to provide a correlation over a broader accuracy spectrum.
> > > >
> > > > **To further illustrate this point, we have now updated our "Stability of Dextr" ablation study in Section 4.3.1 and Figure 2 where architectures are sampled more uniformly across the full accuracy range.** In this revised figure, we observe that Dextr correctly ranks all networks in accordance with their test accuracies, demonstrating that its ranking ability is more clearly revealed when the sampled architectures span a wider performance range.
> > > >
> > > >
> > > > **References**
> > > >
> > > > [1] Poole, Ben, et al. "Exponential expressivity in deep neural networks through transient chaos." Advances in neural information processing systems 29 (2016).
> > > >
> > > > [2] Chen, Wuyang, et al. ‘No Free Lunch in Neural Architectures? A Joint Analysis of Expressivity, Convergence, and Generalization’. AutoML Conference 2023, 2023.

---

> > > > > ### Comment · Reviewer_qJ1j · 2025-07-24
> > > > > **Mostly Satisfied - One More Thing**
> > > > >
> > > > > The reviewer thanks the authors for their efforts in adding more citations and improving Figure 2.
> > > > >
> > > > > For clarity, the reviewer asks the authors to revise the third paragraph of the Introduction (begins with "Another limitation of most", but the blue text) that talks about expressivity to better align with the visual examples in the appendix, e.g., not implying that wide architectures are a weakness.

---

> > > > > > ### Author Response · Authors · 2025-07-24
> > > > > > **Response by Authors**
> > > > > >
> > > > > > The authors appreciate the prompt feedback from the reviewer. In response, **we have revised the third paragraph of the Introduction** to better align with the visual examples provided in the Appendix, and to avoid implying that wide architectures are a weakness.
> > > > > >
> > > > > > The authors once again thank the reviewer for their thorough and very thoughtful review!

---

### Review · Reviewer_o4YZ · 2025-06-06

**Summary Of Contributions:**

The authors introduce a novel score for zero-shot neural architecture search (NAS) that aims to balance generalizability, convergence, and expressivity of neural networks. To construct their score, they exploit that the minimum eigenvalue of the gram matrix of a neural network w.r.t. its input data correlates with convergence and generalizability of a neural network. Additionally, they make use of measuring the output curvature of a neural network using first-order and second-order gradient information with respect to the network input.
In the experiments, it is shown that the proposed score (Dextr) exhibits better Spearman correlation with architecture performances than existing scores, which leads to better results in NAS on two tasks they consider.

**Audience:**

Yes

**Claims And Evidence:**

Yes

**Requested Changes:**

The authors should provide more intuition on Eqs. 3, 4, and 5 (see above) and consider adding a more detailed discussion on the generalization of their theory to CNNs and Transformers. This seems to be important since their experimental evaluation is fully based on CNN and ViT architectures, while their theory only considers simple MLPs, introducing a significant gap between theoretical insights and experimental findings.

To increase readability, the authors should consider incorporating my suggestions in **Minor Weaknesses/Questions** above.

**Strengths And Weaknesses:**

**Strenghts**
- Important and interesting research on zero-shot proxies for NAS, which helps to decrease the cost of NAS methods
- It is fully reasonable to construct a zero-shot proxy that incorporates measures of expressivity, generalizability, and convergence of a neural network
- The experimental setup is reasonable and shows that the proposed score improves upon existing zero-shot and one-shot NAS baselines
- Generally well-structured paper
- Proofs of Theorems given in the Appendix

**Weaknesses**
- It seems that the theoretical motivation of the proposed score only provably holds for small ReLU neural networks with one hidden layer
- It would help to provide an intuition on Eq. 3, 4 & 5, giving the reader an idea why these scores measure convergence, generalizability, and expressivity reliably (can also be given in the Appendix)
- Similarly, Remark 3 should be discussed in more detail. The authors claim that the results from Sec. 3.2.1 and Sec. 3.2.2 can be transferred to CNNs (at least approximately) without giving a rigorous theoretical grounding. Since in CNNs, weight sharing is an integral part of the architecture, which is not reflected in the theory in Sec. 3.2.1 and Sec. 3.2.2. This must be clarified
- In Eq. 8, the authors define their Dextr score, which assumes that one can sum over the inverse of feature condition numbers (which are related to the Gram matrix as shown). However, the theory only deals with MLPs with one hidden layer, making a simple summation questionable in this setting: Since the feature condition number of layer $l$ depends on the feature condition number of layer $l-1$, and since summing over these values assumes independence, I'm not sure if this summation can be reliably done.
- Akin to the generalization of their score/theory to CNNs, the authors argue that - since under some conditions Transformers behave like CNNs - one can also transfer their score and theory to Transformer architectures. While in general I agree that this is possible under some conditions, I think the authors should discuss this in more detail: As they correctly say, Transformers only behave like CNNs if, e.g., the number of attention heads is set carefully, this might not hold in Transformer search spaces used in practice.
- In Sec. 4, an additional ablation would be nice to show that Dextr indeed captures convergence, generalizability and expressivity properties of neural networks.

**Minor Weaknesses/Questions**
- Assumption 1 is not really an assumption, but a Lemma.
- Tab. 1 should be moved into Sec. 4
- in Tab. 4, no standard deviation is shown for the baseline scores. Also, the std is only computed using 3 runs, this should be increased (also given that computing these scores should be cheap).
- Why did the authors decide not to compare their zero-shot method to the baselines on the benchmarks they have used for the correlation experiments?
Tab. 4 should be reported as a plot to increase readability
- Tab. 5 can be merged with Tab. 1
- Sec. 4.3.3: How do the authors explain that the Dextr peaks at layers expanding the dimensionality?

---

> ### Author Response · Authors · 2025-07-10
> **Response to Reviewer o4YZ (1/3)**
>
> We are thankful for the reviewer for their thorough review and helpful comments. Below we address the main concerns of the reviewer.
>
> > It seems that the theoretical motivation of the proposed score only provably holds for small ReLU neural networks with one hidden layer
>
> The reviewer is completely right that the provided theoretical foundation considers small ReLU networks with one hidden layer. However, the covergence theorems developed in [1] and in our paper hold for deep neural networks as well.
>
> Consider a deep neural network of the form
>
> $f(x,\mathbf{W},\mathbf{a})=\mathbf{a}^T\sigma(\mathbf{W}^{(H)}\ldots\sigma(\mathbf{W}^{(1)}\mathbf{x}))$,
>
> where $\mathbf{x}\in\mathbb{R}^d$ is the input and $\mathbf{a}\in \mathbb{R}^m$ is the output layer [1]. If $h=2,\ldots,H$ and  $\mathbf{W}^{(h)}\in \mathbb{R}^{m\times m}$ are the middle layers, then the associated gram matrix $\mathbf{G}^{(h)}(t)\in \mathbb{R}^{n\times n}$ will have values  $\mathbf{G}_{ij}^{(h)}(t) = \left\langle \frac{\partial u_i(t)}{\partial \mathbf{W}^{(h)}(t)}, \frac{\partial u_j(t)}{\partial \mathbf{W}^{(h)}(t)} \right\rangle$, where $u_i$ and $u_j$ are the i-th and j-th predictions respectively.
>
> Du et al. [1] show that $\sum_{h=1}^{H} \mathbf{G}^{(h)}(t)$ has a lower-bounded minimum eigenvalue because $u_i$ and $u_j$ are not parallel. Moreover, $\sum_{h=1}^{H} \mathbf{G}^{(h)}(0)$ is close to the fixed matrix $\sum_{h=1}^{H} \mathbf{G}\infty^{(h)}$, and $\sum_{h=1}^{H} \mathbf{G}^{(h)}(t)$ remains close to $\sum_{h=1}^{H} \mathbf{G}^{(h)}(0)$ throughout training. Therefore, the proof of the convergence theorem in [1] extends naturally to deep neural networks with $H$ hidden layers. In this case, the rate of convergence is governed by $\lambda_{\min}\left( \sum_{h=1}^{H} \mathbf{G}_\infty^{(h)} \right)$. Consequently, Theorems 1 and 3 in our paper are also applicable to deep networks with multiple hidden layers. **We have added this explanation of the generalisability of our convergence theorems to deep neural networks in Appendix Section A.2 of the updated manuscript.**
>
> > It would help to provide an intuition on Eq. 3, 4 & 5, giving the reader an idea why these scores measure convergence, generalizability, and expressivity reliably (can also be given in the Appendix)
>
> **Convergence**
>
>  Consider Eq. 3 from the paper.
>
> $$
> \|\mathbf{u}_i(t)-\mathbf{y}_i\|_2^2 \le \exp(-\lambda{\mathrm{min}}(\mathbf{H}^\infty) t)\|\mathbf{u}_i(0)-\mathbf{y}_i\|_2^2.
> $$
>
> Here, the training loss at initialisation and time $t$ correspond to the terms $|\mathbf{u}_i(0) - \mathbf{y}_i|_2^2$ and $|\mathbf{u}_i(t) - \mathbf{y}_i|_2^2$ respectively. We can observe that the term $\exp\left(-\lambda{\mathrm{min}}(\mathbf{H}^\infty) t\right)$ is multiplied to the training loss at initialisation, and governs the upper bound of training loss at time $t$. If we take the training loss at initialisation out of consideration, then higher the term $\lambda{\mathrm{min}}(\mathbf{H}^\infty)$ would be, lower the term $\exp\left(-\lambda{\mathrm{min}}(\mathbf{H}^\infty) t\right)$ would be, and lower the upper bound of the training loss at time $t$, $|\mathbf{u}_i(t) - \mathbf{y}_i|2^2$, would be, which shows better convergence. In conclusion, the term $\lambda{\mathrm{min}}(\mathbf{H}^\infty)$ at a given time $t$ controls the upper bound of the training loss at $t$ and hence, governs the convergence rate.
>
> **Generalisation**
>
> Consider Eq. 4 from the paper:
>
> $$
> \mathbb{E}[L(\mathbf{W})] \le \mathcal{O}\left(C_2 \sqrt{\frac{\mathbf{y}^\top \mathbf{y}}{\lambda_{\min}(\mathbf{H}^\infty)\, N}}\right) + \mathcal{O}\left(\sqrt{\frac{\log(1/\delta)}{N}}\right).
> $$
>
> We observe that the upper bound of the expected test loss (\$\mathbb{E}\[L(\mathbf{W})]\$) is governed by two terms. The smaller the term \$\mathcal{O}\left(C\_2 \sqrt{\frac{\mathbf{y}^\top \mathbf{y}}{\lambda\_{\min}(\mathbf{H}^\infty), N}}\right)\$, the lower the upper bound on the test loss. Therefore, a larger value of \$\lambda\_{\min}(\mathbf{H}^\infty)\$ leads to a lower test loss and, consequently, better generalisation.
>
> **Expressivity**
>
> The main intuition behind our expressivity metric is that extrinsic curvature of the output \$\kappa(\theta)\$ captures how sensitive the network's output is to small changes in its input \[2]. This property is directly connected to the expressivity as an expressive model can adapt finely to differences between input samples, which is reflected in high local curvature.
> **We have refined our explanation on Convergence, Generalisation and Expressivity in Section 3.2. of the updated manuscript.**

---

> ### Author Response · Authors · 2025-07-10
> **Response to Reviewer o4YZ (2/3)**
>
> > Similarly, Remark 3 should be discussed in more detail. The authors claim that the results from Sec. 3.2.1 and Sec. 3.2.2 can be transferred to CNNs (at least approximately) without giving a rigorous theoretical grounding. Since in CNNs, weight sharing is an integral part of the architecture, which is not reflected in the theory in Sec. 3.2.1 and Sec. 3.2.2. This must be clarified
>
> We thank the reviewer for the suggestion. Below we explain how convolutional layers can be mathematically represented and then approximated as an over-parameterized neural network layer, following the derivation from [3].
>
> Jiang et. al [3] show that the convolutional layer operation with an input $x_{in} \in \mathbb{R}^{c_{in} \times w \times h}$ and a filter $w \in \mathbb{R}^{c_{in} \times k \times k}$, with a stride of one and zero padding, can formally be expressed as:
>
> $[x_{out}]_{i,j}=\sum_{c=1}^{c_{in}}\sum_{a=-p}^{p}\sum_{b=-p}^{p}[w^{c}]_{a+p+1,b+p+1}\cdot[x_{in}^{c}]_{a+i,b+j}$
>
> where $x_{in}^{c}$ and $w^{c}$ represent the c-th channel of $x_{in}$ and $w$ respectively, $p=(k-1)/2$, and $i\in[w]$, $j\in[h]$.
>
> Next, $x_{out}$ and each input channel $x_{in}^{c}$ can be flattened into one-dimensional vectors, $\tilde{x}_{out}\in\mathbb{R}^{d\times1}$ and $\tilde{x}_{in}^{c}\in\mathbb{R}^{d\times1}$ respectively, where $d=w\times h$. This transformation leads to the following approximation, viewing the convolutional layer as a multi-sample fully-connected network:
>
> $\tilde{x}_{out}=\sum_{c=1}^{c_{in}}A_{c}\sigma((B_{c}\circ W)^{T}\tilde{x}_{in}^{c})$
>
> subject to the constraints:
>
> $A_{c}\in\mathbb{R}^{d\times d_{h}}$ with $[A_{c}]_{ij}=\mathbb{I}\{j=(c-1)d+i\}$
> $B_{c}\in\mathbb{R}^{d_{h}\times d}$ with $[B_{c}]_{ij}=\mathbb{I}\{(c-1)d<i\le cd\}$, and
> $B_c\circ W$ satisfying weight sharing constraints.
>
> Here, $d_{h}=c_{in}\times d$ and $W\in\mathbb{R}^{d\times d_{h}}$ is the weight matrix. We can further simplify this by relaxing the second constraint to $B_c=1_{d_h \times d}$ and ignore the last constraint. As a result, we view each flattened input channel $\tilde{x}_{in}^{c}$ as an independent data sample, forming a multi-sample fully-connected network. **We have included a detailed explanation of the transferability of our theory to CNNs in the Appendix Section A.3 of our updated manuscript.**
>
> > In Eq. 8, the authors define their Dextr score, which assumes that one can sum over the inverse of feature condition numbers (which are related to the Gram matrix as shown). However, the theory only deals with MLPs with one hidden layer, making a simple summation questionable in this setting: Since the feature condition number of layer depends on the feature condition number of layer l-1, and since summing over these values assumes independence, I'm not sure if this summation can be reliably done.
>
> We are thankful for the reviewer for raising this critical point. In a previous response to a weakness by the reviewer, we showed how Theorems 1 and 3 are transferable to deep neural networks with $H$ hidden layers and that $\lambda_{min}(\sum_{h=1}^{H}\mathbf{G}_\infty^{(h)})$ would then govern the convergence rate of a network with $H$ layers.
>
> Using Weyl's inequality for Hermitian matrices, we know that
>
> $\sum_{h=1}^{H} \lambda_{min}\mathbf{G}_\infty^{(h)} \leq\lambda_{min}(\sum_{h=1}^{H} \mathbf{G}_\infty^{(h)})$.
>
> Through this inequality, we can observe that the sum of individual minimum eigenvalues of each layer’s Gram matrix provides a lower bound on the total minimum eigenvalue governing convergence. Adding negative and exponential on both sides, we know that
>
> $-\exp(\sum_{h=1}^{H} \lambda_{min}\mathbf{G}_\infty^{(h)}) \geq-\exp(\lambda_{min}(\sum_{h=1}^{H} \mathbf{G}_\infty^{(h)}))$.
>
> Hence,  $\sum_{h=1}^{H} \lambda_{min}\mathbf{G}_\infty^{(h)}$  can be replaced with $\lambda_{min}(\sum_{h=1}^{H} \mathbf{G}_\infty^{(h)})$ in the convergence theorem for multi-layer networks. Hence,  $\sum_{h=1}^{H} \lambda_{min}\mathbf{G}_\infty^{(h)}$, utilised by Dextr, also governs the convergence rate of a deep neural network.  **We have included this explanation of the generalisability to deep neural networks in the Appendix Section A.2 of our updated manuscript.**

---

> > ### Author Response · Authors · 2025-07-10
> > **Formatting fix for Response (2/3)**
> >
> > **Since our previous Response (2/3) has some formatting issues, here is the corrected version of the response. Please ignore the previous Response (2/3). We apologise for the inconvenience.**
> >
> > > Similarly, Remark 3 should be discussed in more detail. The authors claim that the results from Sec. 3.2.1 and Sec. 3.2.2 can be transferred to CNNs (at least approximately) without giving a rigorous theoretical grounding. Since in CNNs, weight sharing is an integral part of the architecture, which is not reflected in the theory in Sec. 3.2.1 and Sec. 3.2.2. This must be clarified
> >
> > We thank the reviewer for the suggestion. Below we explain how convolutional layers can be mathematically represented and then approximated as an over-parameterized neural network layer, following the derivation from [3].
> >
> > Jiang et. al [3] show that the convolutional layer operation with an input $x_{in} \in \mathbb{R}^{c_{in} \times w \times h}$ and a filter $w \in \mathbb{R}^{c_{in} \times k \times k}$, with a stride of one and zero padding, can formally be expressed as:
> >
> > $[x\_{out}]\_{i,j}=\sum\_{c=1}^{c\_{in}}\sum\_{a=-p}^{p}\sum\_{b=-p}^{p}[w^{c}]\_{a+p+1,b+p+1}\cdot[x\_{in}^{c}]\_{a+i,b+j}$
> >
> > where $x_{in}^{c}$ and $w^{c}$ represent the c-th channel of $x_{in}$ and $w$ respectively, $p=(k-1)/2$, and $i\in[w]$, $j\in[h]$.
> >
> > Next, $x_{out}$ and each input channel $x_{in}^{c}$ can be flattened into one-dimensional vectors, $\tilde{x}\_{out}\in\mathbb{R}^{d\times1}$ and $\tilde{x}\_{in}^{c}\in\mathbb{R}^{d\times1}$ respectively, where $d=w\times h$. This transformation leads to the following approximation, viewing the convolutional layer as a multi-sample fully-connected network:
> >
> > $\tilde{x}\_{out}=\sum\_{c=1}^{c\_{in}}A\_{c}\sigma((B\_{c}\circ W)^{T}\tilde{x}\_{in}^{c})$
> >
> > subject to the constraints:
> >
> > $A\_{c}\in\mathbb{R}^{d\times d\_{h}}$ with $[A\_{c}]\_{ij}=\mathbb{I}\{j=(c-1)d+i\}$
> >
> > $B\_{c}\in\mathbb{R}^{d\_{h}\times d}$ with $[B\_{c}]\_{ij}=\mathbb{I}\{(c-1)d<i\le cd\}$, and
> >
> > $B\_c\circ W$ satisfying weight sharing constraints.
> >
> > Here, $d_{h}=c_{in}\times d$ and $W\in\mathbb{R}^{d\times d_{h}}$ is the weight matrix. We can further simplify this by relaxing the second constraint to $B_c=1_{d_h \times d}$ and ignore the last constraint. As a result, we view each flattened input channel $\tilde{x}_{in}^{c}$ as an independent data sample, forming a multi-sample fully-connected network. **We have included a detailed explanation of the transferability of our theory to CNNs in the Appendix Section A.3 of our updated manuscript.**
> >
> > > In Eq. 8, the authors define their Dextr score, which assumes that one can sum over the inverse of feature condition numbers (which are related to the Gram matrix as shown). However, the theory only deals with MLPs with one hidden layer, making a simple summation questionable in this setting: Since the feature condition number of layer depends on the feature condition number of layer l-1, and since summing over these values assumes independence, I'm not sure if this summation can be reliably done.
> >
> > We are thankful for the reviewer for raising this critical point. In a previous response to a weakness by the reviewer, we showed how Theorems 1 and 3 are transferable to deep neural networks with $H$ hidden layers and that $\lambda_{min}(\sum_{h=1}^{H}\mathbf{G}_\infty^{(h)})$ would then govern the convergence rate of a network with $H$ layers.
> >
> > Using Weyl's inequality for Hermitian matrices, we know that
> >
> > $\sum\_{h=1}^{H} \lambda\_{min}\mathbf{G}\_\infty^{(h)} \leq\lambda\_\{min}(\sum\_{h=1}^{H} \mathbf{G}\_\infty^{(h)})$.
> >
> > Through this inequality, we can observe that the sum of individual minimum eigenvalues of each layer’s Gram matrix provides a lower bound on the total minimum eigenvalue governing convergence. Adding negative and exponential on both sides, we know that
> >
> > $-\exp(\sum\_{h=1}^{H} \lambda\_{min}\mathbf{G}\_\infty^{(h)}) \geq-\exp(\lambda\_{min}(\sum\_{h=1}^{H} \mathbf{G}\_\infty^{(h)}))$.
> >
> > Hence,  $\sum\_{h=1}^{H} \lambda\_{min}\mathbf{G}\_\infty^{(h)}$  can be replaced with $\lambda\_{min}(\sum\_{h=1}^{H} \mathbf{G}\_\infty^{(h)})$ in the convergence theorem for multi-layer networks. Hence,  $\sum\_{h=1}^{H} \lambda\_{min}\mathbf{G}\_\infty^{(h)}$, utilised by Dextr, also governs the convergence rate of a deep neural network.  **We have included this explanation of the generalisability to deep neural networks in the Appendix Section A.2 of our updated manuscript.**

---

> ### Author Response · Authors · 2025-07-10
> **Response to the Reviewer o4YZ (3/3)**
>
> > Akin to the generalization of their score/theory to CNNs, the authors argue that - since under some conditions Transformers behave like CNNs - one can also transfer their score and theory to Transformer architectures. While in general I agree that this is possible under some conditions, I think the authors should discuss this in more detail: As they correctly say, Transformers only behave like CNNs if, e.g., the number of attention heads is set carefully, this might not hold in Transformer search spaces used in practice.
>
> We agree to the reviewers concern that in some cases, strict equivalence of ViTs and CNNs is not met, specifically, in the cases of practical ViT search spaces (like AutoFormer) where relative positional encodings are not always used, or where the number of attention heads may not be large enough to achieve the convolutional limit. However, we argue that the reason why Dextr is generalisable on ViTs is because a strict equivalence is not necessary for Dextr to be effective. As long as the output of the transformer layers exhibit sufficient structure (just like in CNNs), the SVD analysis is meaningful. Importantly, our theoretical derivation interprets inputs or features as collections of vectors (channels or tokens), on which SVD analysis is performed. Notably, the central assumption of our theory is not convolution, but rather the existence of a meaningful Gram matrix formed from activations. Since ViT block outputs (after Multi-head Self Attention or MLP) can be reshaped or projected into such vector sets, our analysis of convergence and generalisation via Gram eigenvalues, and hence SVD, extends naturally. **We have elaborated more on application of Dextr to ViTs and limting cases in Section 3.3.4 of our updated manuscript.**
>
> > Why did the authors decide not to compare their zero-shot method to the baselines on the benchmarks they have used for the correlation experiments?
>
> In our NAS experiments, reported in Table 2 and 3, we have utilised the evaluation protocol used in the most recent state-of-the-art papers like MeCo [3] and ZiCo [4] for fair comparison with these methods. Since older zero-shot NAS methods like Grasp [5], Fisher [5], Synflow [5], etc. use a different evaluation protocol in their NAS experiments, our numbers are not directly comparable with theirs.
>
> > Sec. 4.3.3: How do the authors explain that the Dextr peaks at layers expanding the dimensionality?
>
> We observe that the inverse condition number, i.e., feature linear independence, peaks at layers where the network expands its channel dimensionality because dimensionality expansion increases the rank capacity of the feature matrix, allowing it to span a larger subspace with more orthogonal directions. As a result, the collinearity between feature channels is reduced, and feature linear independence (FMI) peaks. Later, when the network compresses feature dimensionality, the rank of the feature matrix decreases, which squeezes the singular value spectrum and FMI drops. **We have included this explanation in Section 4.3.3 of our updated manuscript.**
>
>
> ### Other Adjustments in the updated manuscript
> - Assumption 1 changed to Lemma 1
> - Repositioned Table 1
> - Table 4 now reported as a plot in Figure 2.
>
> **References**
>
> [1] Du, Simon S., et al. "Gradient Descent Provably Optimizes Over-parameterized Neural Networks." International Conference on Learning Representations. 2018.
>
> [2] Poole, Ben, et al. "Exponential expressivity in deep neural networks through transient chaos." Advances in neural information processing systems 29 (2016).
>
> [3] Jiang, Tangyu, Haodi Wang, and Rongfang Bie. "Meco: zero-shot NAS with one data and single forward pass via minimum eigenvalue of correlation." Advances in Neural Information Processing Systems 36 (2023): 61020-61047.
>
> [4] Li, Guihong, et al. "ZiCo: Zero-shot NAS via inverse Coefficient of Variation on Gradients." The Eleventh International Conference on Learning Representations.
>
> [5] Abdelfattah, Mohamed S., et al. "Zero-Cost Proxies for Lightweight NAS." International Conference on Learning Representations.

---

> > ### Comment · Reviewer_o4YZ · 2025-07-17
> >
> > Thank you for the thorough response to my feedback and for the changes made in the manuscript. I think the changes significantly increased the paper's quality.
> >
> > I only have one follow-up remark regarding FMI: I think this behavior does not only depend on the network's structure, but also on the way it is optimized. For example, if the network is encouraged to learn disentangled representations, I'd expect the layer outputs to become more independent. Since zero-shot NAS is not aware of the optimization procedure of the network, it would be interesting to see how well your approach works when the chosen architectures are optimized with different goals/regularizations (e.g., aiming for disentanglement, etc.). Though I understand that this might be out of scope for that paper.

---

> > > ### Author Response · Authors · 2025-07-21
> > > **Response to the Official Comment by Reviewer o4YZ**
> > >
> > > We fully agree to the statement that if a network is encoraged to learn disentangled representations during training (for e.g. through a regulariser), the layer outputs will become more independent. We empirically found through a toy example that if FMI is used as a regulariser in the training process, the FMI indeed increases. However, such a case would require an evaluation of Dextr on a *trained network* unlike an untrained setting that we (and related work) follow. Since our goal is to assess architectural quality at initialisation based solely on structural and geometric properties, we believe that a deeper study on Dextr's behaviour during training is out of scope for this work. However, we agree to the reviewer that it is an interesting future work.

---

### Review · Reviewer_abwc · 2025-06-30

**Summary Of Contributions:**

This paper introduces Dextr, a 'zero-cost proxy' NAS technique that uses unlabeled data to score architectures---this is often used to either generate a ranking over a small search space or to guide a local search in a large search space where architectures cannot easily be enumerated. Most zero-cost proxy techniques either use labeled data, which might not be realistically available, or they do not use data at all, which make such techniques brittle to changes in the target task. The actual technique is motivated by two observations: 1. feature collinearity negatively impacts performance, and 2. network outputs at random initialization that are more curved implicate higher expressivity, perhaps as a result of lower collinearity.

**Audience:**

Yes

**Broader Impact Concerns:**

None.

**Claims And Evidence:**

Yes

**Requested Changes:**

See weaknesses above.

**Strengths And Weaknesses:**

**Strengths**
- Lower collinearity being correlated with higher expressivity and better performance makes a lot of intuitive sense, and the authors do a great job of motivating this particular approach in the introduction.
- Dextr has a clean theoretical motivation that is mostly easy to follow.
- Dextr appears to substantially outperform most existing methods on many of the tasks on which the authors evaluate. Furthermore, the experimental results cover both simplified NAS settings where the entire search space can be evaluated and ranked explicitly (e.g. NAS-Bench-201), as well as more realistic settings with much larger search spaces (e.g. DARTS).

**Weaknesses**
- It's not immediately clear to me why collinearity would arise as a property of the architecture. Rather, I would think that this would instead be a property of poor initialization. Can the authors elaborate on their intuition for this, or perhaps provide a motivating example of an architecture that is prone to collinearity?
- While not crucial to the evaluation, it would be useful to see how Dextr performs across different tasks using the same search space but with wildly different final rankings. For example, Figure 3 in NAS-Bench-360 [1] shows that while NAS-Bench-201 rankings are mostly correlated across vision tasks, they become completely uncorrelated when comparing CIFAR-100 to prosthetics control and PDE solving tasks. Naturally, zero-cost proxies that are data agnostic would completely fail in such settings, which is discussed in [2]. It makes intuitive sense that unlabeled data usage should overcome this limitation of data agnostic proxies, but making this more explicit either with an additional discussion or experiment would make the choice to use unlabeled data more compelling. With that said, this isn't a central claim of the paper, so I don't view this as crucial to the evaluation.
- Table 5 shows a comparison between Dextr and the harmonic mean between the Zen (which focuses on expressivity) and ZiCo methods (which focuses on convergence and generalization). The reason for the comparison is that Dextr, on the other hand, balances expressivity, convergence, and generalization roughly by taking the harmonic mean of a curvature term and an inverse condition number term. What seems missing from this table are the individual performance values for Zen and ZiCo on their own---does the harmonic mean somehow make performance worse in the case of these proxies? For that matter, it would also be useful to understand how Zen compares to just using the curvature term in Dextr (both expressivity focused), and how ZiCo compares to just using the inverse condition number term (with both focusing on convergence/generalization).

[1] https://arxiv.org/abs/2110.05668
[2] https://iclr-blog-track.github.io/2022/03/25/zero-cost-proxies/

---

> ### Author Response · Authors · 2025-07-10
> **Response to reviewer abwc (1/2)**
>
> We appreciate the reviewer's thorough evaluation and insightful comments. Below, we address the concerns raised.
>
> > It's not immediately clear to me why collinearity would arise as a property of the architecture. Rather, I would think that this would instead be a property of poor initialization. Can the authors elaborate on their intuition for this, or perhaps provide a motivating example of an architecture that is prone to collinearity?
>
> We thank the reviewer for raising this point. We argue that collinearity arises from both initialization and architectural inductive biases. However, initialisation plays a less important role. This is because while weight initialization affects the specific values of features, the architecture determines the degree to which the network processes input redundantly, and hence how likely the outputs are to be collinear. For example, architectures with limited width or deep networks with repeated identical blocks are structurally constrained in their ability to decorrelate inputs. Therefore, the outputs would have high colinearity. To mitigate the effect of initialisation even further, our correlation experiments are conducted for multiple runs with different initialisations. We also empirically confirm the neglegible effect of initialisation in our stability analysis (Figure 2), where Dextr (and hence collinearity) remains stable across different seeds with different random initialisations. **We provide a motivating example of an architecture prone to colinearity in the introduction of our updated manuscript.**
>
> > While not crucial to the evaluation, it would be useful to see how Dextr performs across different tasks using the same search space but with wildly different final rankings...
>
> We appreciate the reviewer highlighting this important point. We agree that evaluating zero-cost proxies across tasks with divergent final rankings is crucial for demonstrating robustness, especially in contrast to data-agnostic proxies which are known to fail under such settings.
>
> In fact, we already address this in our experiments on TransNAS-Bench-101-micro (TNB101) [2], which is specifically designed to evaluate NAS methods across heterogeneous tasks using the same search space. The tasks chosen from TransNAS-Bench-101 have signifincantly low correlation between final rankings of architectures, especially when considering top 50\% of the architectures as observed in Figure 4 of [2]. This demonstrates the versatility of the benchmark. Despite this, Dextr consistently achieves strong correlations across all three tasks, significantly outperforming both data-agnostic proxies (e.g., #params, Synflow) and even other data-dependent ones. This provides empirical evidence that Dextr’s use of unlabeled input data helps it remain robust to task variations, as the reviewer anticipated. **We have included this explanation regarding our results on TransNAS-Bench-101 in Section 4.1 of our updated manuscript.**

---

> ### Author Response · Authors · 2025-07-10
> **Response to Reviewer abwc (2/2)**
>
> > What seems missing from Table 5 are the individual performance values for Zen and ZiCo on their own---does the harmonic mean somehow make performance worse in the case of these proxies? For that matter, it would also be useful to understand how Zen compares to just using the curvature term in Dextr (both expressivity focused), and how ZiCo compares to just using the inverse condition number term (with both focusing on convergence/generalization).
>
> We thank the reviewer for this suggestion. Now, in our ablation study, we additionally demonstrate the importance of individual characteristics in Dextr, namely the condition number $c(\mathbf{X})$ and the extrinsic curvature $\kappa$ and compare them with ZiCo and Zen (proxies focused on convergence/generalisation and expressivity respectively). We can observe the updated results below.
>
> Table 1: Comparison of Spearman Rank Correlation Coefficient (ρ) using individual characteristics with Zen and ZiCo on NAS-Bench-201, along with the comparison of Dextr with the harmonic mean of Zen and ZiCo. Here, C/G refers to convergence/generalisation and E refers to expressivity metric.
>
> | Component                     | Property Type | CIFAR-10 | CIFAR-100 | ImageNet16-120 |
> |------------------------------|----------------|----------|------------|----------------|
> | $\log(1 + c(\mathbf{X}))$    | C/G            | **0.88**     | **0.89**       | **0.85**           |
> | ZiCo   | C/G            | 0.76     | 0.79       | 0.77           |
> | $\log(1 + \kappa)$           | E              | **0.51**     | **0.49**       | **0.51**           |
> | Zen           | E              | 0.33     | 0.34       | 0.38           |
> | HM(Zen,ZiCo)           | C/G+E              | 0.44     | 0.44       | 0.47           |
> | **Dextr (combined)**         | C/G + E        | **0.90** | **0.91**   | **0.87**       |
>
>
> The results demonstrate that Dextr, which combines convergence/generalisation and expressivity components, consistently achieves the highest Spearman rank correlation across all three datasets in NAS-Bench-201. Notably, the condition number term $\log(1 + c(\mathbf{X}))$ outperforms ZiCo across all tasks, indicating that Dextr’s convergence/generalisation measure is a more reliable proxy for performance than ZiCo. Moreover, the curvature term $\log(1 + \kappa)$ also consistently outperforms Zen, indicating its effectiveness. Lastly, the harmonic mean of Zen and ZiCo performs worse than either of Dextr’s individual components and significantly worse than Dextr itself, suggesting that naively combining external proxies is not as effective as Dextr. **We have updated our ablation study in the Section 4.3.2 with the above experiment.**
>
> **References**
>
> [1] Chen, Wuyang, et al. ‘No Free Lunch in Neural Architectures? A Joint Analysis of Expressivity, Convergence, and Generalization’. AutoML Conference 2023, 2023.
> [2] Duan, Yawen, et al. "Transnas-bench-101: Improving transferability and generalizability of cross-task neural architecture search." Proceedings of the IEEE/CVF Conference on Computer Vision and Pattern Recognition. 2021.

---

### Author Response · Authors · 2025-07-10
**Thank you!**

**Dear Action Editor and Reviewers,**

We would like to extend our heartfelt thanks to the Action Editor and reviewers **abwc**, **o4YZ**, and **qJ1j** for their time, thoughtful feedback, and constructive suggestions. We deeply appreciate their efforts in evaluating our work and providing valuable insights.

In addition to the encouraging comments, the reviewers raised important questions and concerns that helped us clarify and immensely strengthen our manuscript. We have addressed these points thoroughly in our responses and incorporated the necessary changes into the revised manuscript. For clarity, all additions have been highlighted in blue.

Please let us know if you have any remaining concerns and thank you once again for your careful reading and thoughtful evaluation of our work.

Sincerely,

Dextr Authors

---

### Author Response · Authors · 2025-08-19
**Camera Ready Version**

Dear Action Editor and Reviewers,

We have now uploaded the camera-ready version. We would like to extend our sincere gratitude to the reviewers and the action editor again for their input on this paper.


Sincerely,

Dextr Authors

---

### Decision · Action_Editor_peZ1 · 2025-08-15

**Recommendation:** Accept as is

**Additional Comments:**

This is a solid paper within neural architecture search. It builds zero-cost proxies for scenarios where labeled data is unavailable while balancing multiple desirable properties.

Reviewers agree on acceptance; the only requests were for further clarifying a number items. The authors' updated version is much more clear and is ready for acceptance.

**Audience:**

Yes

**Audience Explanation:**

Yes, neural architecture search is a core area and is within the TMLR audience's interests.

**Claims And Evidence:**

Yes

**Claims Explanation:**

This paper has sufficient theoretical and empirical analysis for its claims.